# Transfer of dysbiotic gut microbiota has beneficial effects on host liver metabolism

Simon Nicolas[1,2], Vincent Blasco-Baque[1,2,3] (iD), Audren Fournel[4,5], Jerome Gilleron[6], Pascale Klopp[1,2], Aurelie Waget[1,2], Franck Ceppo[6], Alysson Marlin[4,5], Roshan Padmanabhan[1,2] (iD), Jason S Iacovoni[1,2], François Tercé[1,2], Patrice D Cani[7] (iD), Jean-François Tanti[6], Remy Burcelin[1,2], Claude Knauf[4,5], Mireille Cormont[6] & Matteo Serino[1,2,*,†] (iD)

## Abstract

Gut microbiota dysbiosis has been implicated in a variety of systemic disorders, notably metabolic diseases including obesity and impaired liver function, but the underlying mechanisms are uncertain. To investigate this question, we transferred caecal microbiota from either obese or lean mice to antibiotic-free, conventional wild-type mice. We found that transferring obese-mouse gut microbiota to mice on normal chow (NC) acutely reduces markers of hepatic gluconeogenesis with decreased hepatic PEPCK activity, compared to non-inoculated mice, a phenotypic trait blunted in conventional NOD2 KO mice. Furthermore, transferring of obese-mouse microbiota changes both the gut microbiota and the microbiome of recipient mice. We also found that transferring obese gut microbiota to NC-fed mice then fed with a high-fat diet (HFD) acutely impacts hepatic metabolism and prevents HFD-increased hepatic gluconeogenesis compared to non-inoculated mice. Moreover, the recipient mice exhibit reduced hepatic PEPCK and G6Pase activity, fed glycaemia and adiposity. Conversely, transfer of lean-mouse microbiota does not affect markers of hepatic gluconeogenesis. Our findings provide a new perspective on gut microbiota dysbiosis, potentially useful to better understand the aetiology of metabolic diseases.

**Keywords** gut microbiota transfer; hepatic glucose production; high-fat diet; metabolic diseases; microbiome
**Subject Categories** Genome-Scale & Integrative Biology; Metabolism Microbiology, Virology & Host Pathogen Interaction
**Mol Syst Biol. (2017) 13: 921**

## Introduction

The intricate ecosystem of intestinal microbes, the gut microbiota, actively participates in several functions of the host, beyond digestion (Shanahan, 2002; Turnbaugh *et al*, 2006; Backhed *et al*, 2007; Velagapudi *et al*, 2010; Reinhardt *et al*, 2012; Serino *et al*, 2012a). The altered proportion and activity of bacterial groups of gut microbiota, named *dysbiosis*, characterizes multiple pathologies (Tomasello *et al*, 2011; Haahtela *et al*, 2013; Serban, 2014), such as type 2 diabetes and obesity (Serino *et al*, 2009; Le Chatelier *et al*, 2013). There is also clear evidence that gut microbiota dysbiosis impacts the liver by promoting hepatic steatosis (Dumas *et al*, 2006; Le Roy *et al*, 2013), a common feature of metabolic syndrome. We reported that susceptibility to diet-induced metabolic diseases is characterized by a particular gut microbiota (Serino *et al*, 2012b). Of note, targeting gut microbiota via dietary treatment (Cani *et al*, 2007), fibres (Serino *et al*, 2012b) or antibiotics (Cani *et al*, 2008; Membrez *et al*, 2008) can restore glucose homoeostasis by reducing metabolic inflammation (Shoelson *et al*, 2006).

Our understanding of the impact of gut microbiota on host metabolism (Shanahan, 2002; Turnbaugh *et al*, 2006; Backhed *et al*, 2007; Reinhardt *et al*, 2012; Serino *et al*, 2012a) is based on the use of axenic mice. These mice enabled the discovery of few molecular mechanisms by which the gut microbiota modulates host metabolism (Backhed *et al*, 2007). It is significant that colonization of axenic mice with gut microbiota from animal models of pathology (i.e. obese mice; Turnbaugh *et al*, 2006) or human stools (Chung *et al*, 2012; Atarashi *et al*, 2013) transferred the related phenotype, suggesting gut microbiota as a putative aetiological factor of that pathology.

1 Institut National de la Santé et de la Recherche Médicale (INSERM), Toulouse, France
2 Unité Mixte de Recherche (UMR) 1048, Institut de Maladies Métaboliques et Cardiovasculaires (I2MC), Université Paul Sabatier (UPS), Toulouse Cedex 4, France
3 Faculté de Chirurgie Dentaire de Toulouse, Université Paul Sabatier, Toulouse Cedex, France
4 Toulouse III, Institut de Recherche en Santé Digestive (IRSD) Team 3, "Intestinal Neuroimmune Interactions" INSERM U1220, Université Paul Sabatier, Toulouse Cedex 3, France
5 European Associated Laboratory NeuroMicrobiota (INSERM/UCL), Bâtiment B – Pavillon Lefebvre, Toulouse Cedex 3, France
6 INSERM Unité 1065/Centre Méditerranéen de Médecine Moléculaire (C3M), Université Côte d'Azur, Nice, France
7 Walloon Excellence in Life Sciences and BIOtechnology (WELBIO), Metabolism and Nutrition Research Group, Louvain Drug Research Institute, Université catholique de Louvain, Brussels, Belgium
*Corresponding author. Tel: +33 5 62 74 45 25; E-mail: matteo.serino@inserm.fr
†Present address: IRSD, Université de Toulouse, INSERM, INRA, ENVT, UPS, Toulouse, France

Lack of microbiota in axenic mice determines both structural and functional alterations such as gut hyper-permeability and atrophy of the immune system (Shanahan, 2002). Therefore, we considered whether the detrimental effects of dysbiotic gut microbiota observed in axenic mice could also be observed in healthy conventional mice. To investigate the role of gut microbiota dysbiosis in the aetiology of metabolic diseases, we inoculated conventional, healthy mice with either dysbiotic gut microbiota from diet-induced and *ob/ob* obese mice or eubiotic gut microbiota from lean mice.

We found that transfer of dysbiotic gut microbiota to conventional mice acutely reduces markers of hepatic gluconeogenesis during normal chow and protects towards high-fat diet-increased markers of hepatic gluconeogenesis and adiposity, together with changes in both gut microbiota and microbiome. Similar metabolic results were obtained when mice were inoculated with a dysbiotic gut microbiota from *ob/ob* mice. Conversely, the transfer of eubiotic gut microbiota slightly affected both the gut microbiota composition and related bacterial metabolic functions of recipient mice, which did not show altered markers of hepatic gluconeogenesis on normal chow.

Our results show that transferring a dysbiotic gut microbiota may benefit the host, proposing to reconsider the role of gut microbiota dysbiosis within the aetiology of metabolic diseases.

## Results

To investigate the metabolic effects of transferring gut microbiota, recipient mice never previously treated with antibiotics were used, since antibiotics have been shown to dampen dysbiosis-induced dysmetabolism (Ellekilde *et al*, 2014) or even to limit the establishment of exogenous microbiota (Manichanh *et al*, 2010).

### Metabolic effects of dysbiotic vs. eubiotic gut microbiota transfer in conventional mice fed a normal chow (NC)

To investigate the role of gut microbiota dysbiosis in the aetiology of metabolic diseases, we transferred the caecal content from high-fat diet-induced obese mice (HFD-microbiota hereafter) into conventional (Conv) mice fed a NC (Conv + OM (HFD); OM stands for "obese microbiota") and we compared this group to mice inoculated with either the vehicle (Conv + PBS) or an eubiotic gut microbiota from lean mice (Conv + LM; LM stands for "lean microbiota"; Fig 1A). For both donor and recipient mice, basal metabolic features are reported in Appendix Fig S1A–E. First, we verified that bacteria from both transplants were viable. We found a decreased amount of cultivable bacteria in the inoculum from obese mice, mainly in the anaerobic bacteria (Appendix Fig S1F and G). Since the majority of gut microbes is not cultivable, we further quantified the DNA content in both transplants. As expected (Daniel *et al*, 2014), the transplants from either obese or lean mice were highly divergent in terms of amount and taxonomy (Appendix Fig S1H and I). By contrast, the two transplants from the same donor showed a strong homogeneity after 1 week (Appendix Fig S1J and K).

With regard to the metabolic impact of both transplants, mice receiving the HFD-microbiota showed a lower 6 h fasting glycaemia when compared to control mice (Fig 1B). The blood glucagon pathway was not significantly affected, as shown by the analysis of hepatic phosphorylation of glucagon PKA targets, together with no change in hepatic glycogen content (Appendix Fig S2A and B). Then, we analysed hepatic gluconeogenesis by performing a pyruvate tolerance test; mice receiving the HFD-microbiota showed a significant lower fasting glycaemia and a concomitant lower hepatic gluconeogenesis compared to control mice, whereas the inoculation with lean microbiota did not induce a significant effect (Fig 1C). Protein level of key hepatic gluconeogenic enzymes PEPCK and G6Pase was not significantly changed (Appendix Fig S2C). By contrast, mice inoculated with the HFD-microbiota showed lower activity for PEPCK (Fig 1D) but not G6Pase (Fig 1E), with no change induced by the lean microbiota. The lower activity for PEPCK could offer a mechanism to explain the lower hepatic gluconeogenesis. Moreover, since the area under the curve shows a not significant HFD-microbiota effect (Fig 1C), also the fasting glycaemia accounts for the observed trend of reduced hepatic gluconeogenesis. Several metabolic parameters were not affected in inoculated mice including body and liver weight, hepatic triglycerides content, liver inflammation, hepatic damage, oral glucose tolerance (Appendix Fig S2D–J) or an index of systemic inflammation analysed by enumerating plasma immune cells (Appendix Fig S3A–D). These data show that the reduction in hepatic gluconeogenesis was not due to hepatic damage.

To explain the reduced fasting glycaemia, we conducted an extensive analysis by microarray to look for overall variations of hepatic gene expression. Out of the totality of genes significantly ($P < 0.05$) modulated (1,021 by the HFD-microbiota vs. Conv + PBS and 1,329 by the lean microbiota vs. Conv + PBS), we identified a network of hepatic metabolic genes whose expression was reduced by the HFD-microbiota (Fig 1F) and involved in *de novo* lipogenesis (Appendix Fig S2K). Among the 1,021 genes significantly modulated by HFD-microbiota none of them was directly implicated in gluconeogenesis, suggesting that the decrease in markers of hepatic glucose production observed above is not due to a change in gene expression.

In mice inoculated with the HFD-microbiota, we also found a serum metabolomic signature of the hepatic phenotype by higher levels of glucogenic precursors, such as lactate and pyruvate (Fig 2A–C), suggesting the reduction in hepatic gluconeogenesis shown in Fig 1C.

These data show that the transfer of HFD-microbiota lowered fasting glycaemia and markers of hepatic gluconeogenesis in association with a reduced gluconeogenic enzyme activity, without affecting neither glucagon signalling nor hepatic glycogen content.

### Analysis of gut barrier in conventional mice fed a NC and inoculated with either a dysbiotic or eubiotic gut microbiota

To explain whether the hepatic phenotype may be dependent on alterations of the gut-to-liver axis (Szabo *et al*, 2010), we analysed the intestinal barrier. First, neither the dysbiotic nor the eubiotic gut microbiota transfer significantly affected the *in vivo* gut permeability (Fig 3A), in accordance with unchanged LPS plasma levels (Fig 3B). Then, since we already showed the impact of gut microbiota dysbiosis on the ileum (Amar *et al*, 2011; Serino *et al*, 2012b), we focused on this intestinal region. Goblet cells and the

    

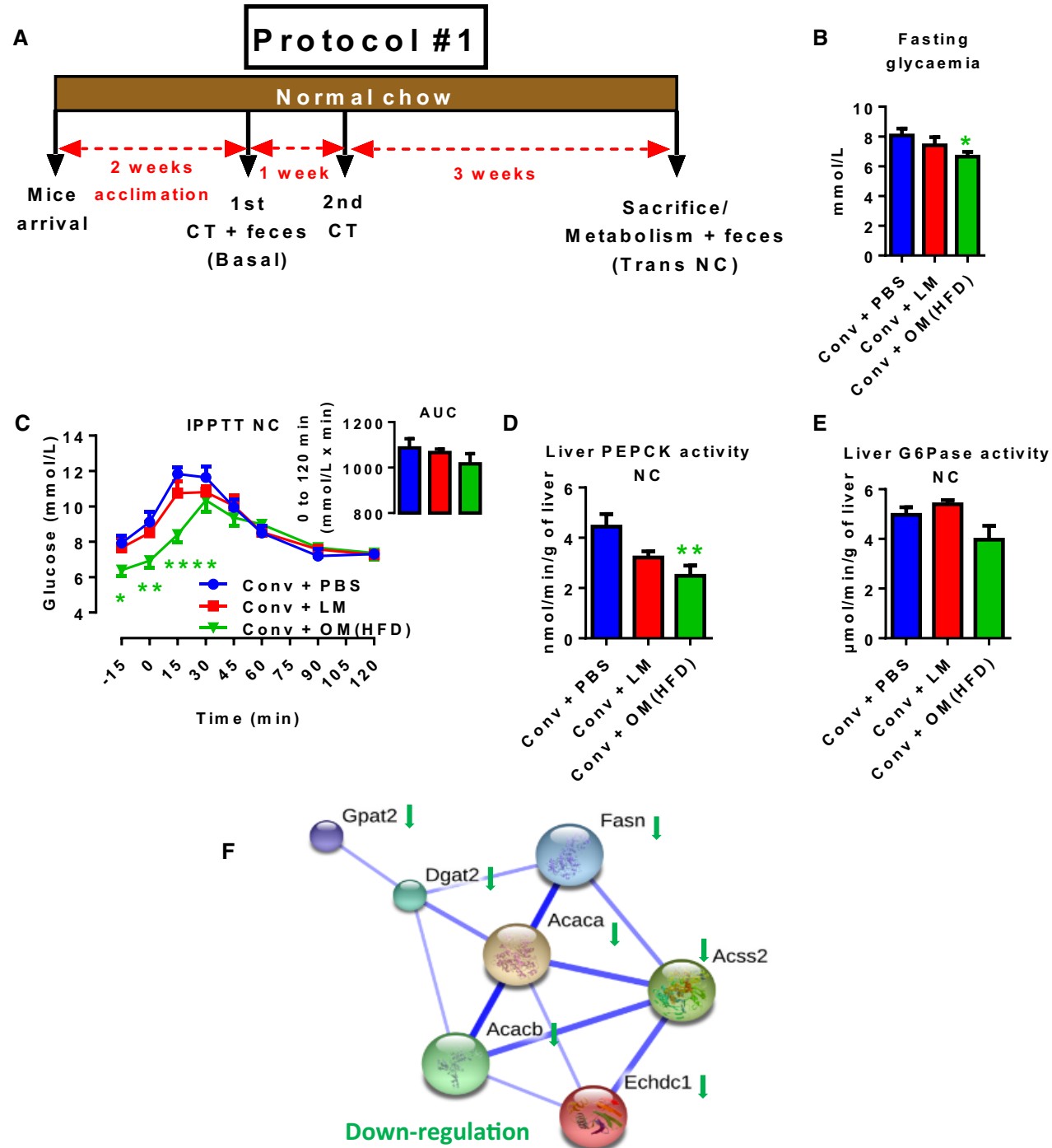

**Figure 1.  Transfer of dysbiotic vs. eubiotic gut microbiota in NC-fed conventional mice reduces hepatic gluconeogenesis.**

A–F  (A) Experimental timeline: 1st/2nd CT (caecal transfer); (B) 6 h fasting glycaemia; (C) intraperitoneal pyruvate tolerance test and AUC as inset; hepatic (D) PEPCK and (E) G6Pase enzymatic activity; (F) String analysis of significantly modulated hepatic metabolic genes analysed by microarray in antibiotic-free NC-fed conventional mice inoculated with either the vehicle (PBS) or caecal microbiota from either lean mice or HFD-fed mice (Conv + PBS, Conv + LM, Conv + OM(HFD), respectively). Data are shown as mean ± SEM; $n = 6$, *$P < 0.05$, **$P < 0.01$, ****$P < 0.0001$; unpaired Student's $t$-test for (B), two-way ANOVA and Sidak's post-test vs. Conv + PBS (C), one-way ANOVA and Dunnett's post-test vs. Conv + PBS (D). Basal, baseline; Trans NC, transfer during NC; Conv, conventional; OM, obese microbiota; LM, lean microbiota.

expression of Muc-2 (the main mucus-producing gene) were used as indices of mucus production and were not significantly affected (Fig 3C and D). No significant modulation of the expression of tight junction genes (Claudin-2/-7, Jam-A, Occludin or ZO-1) was observed (Fig 3E). With regard to inflammation, no significant change was observed for FoxP3, IL-17a, IFNγ and NF-kB gene

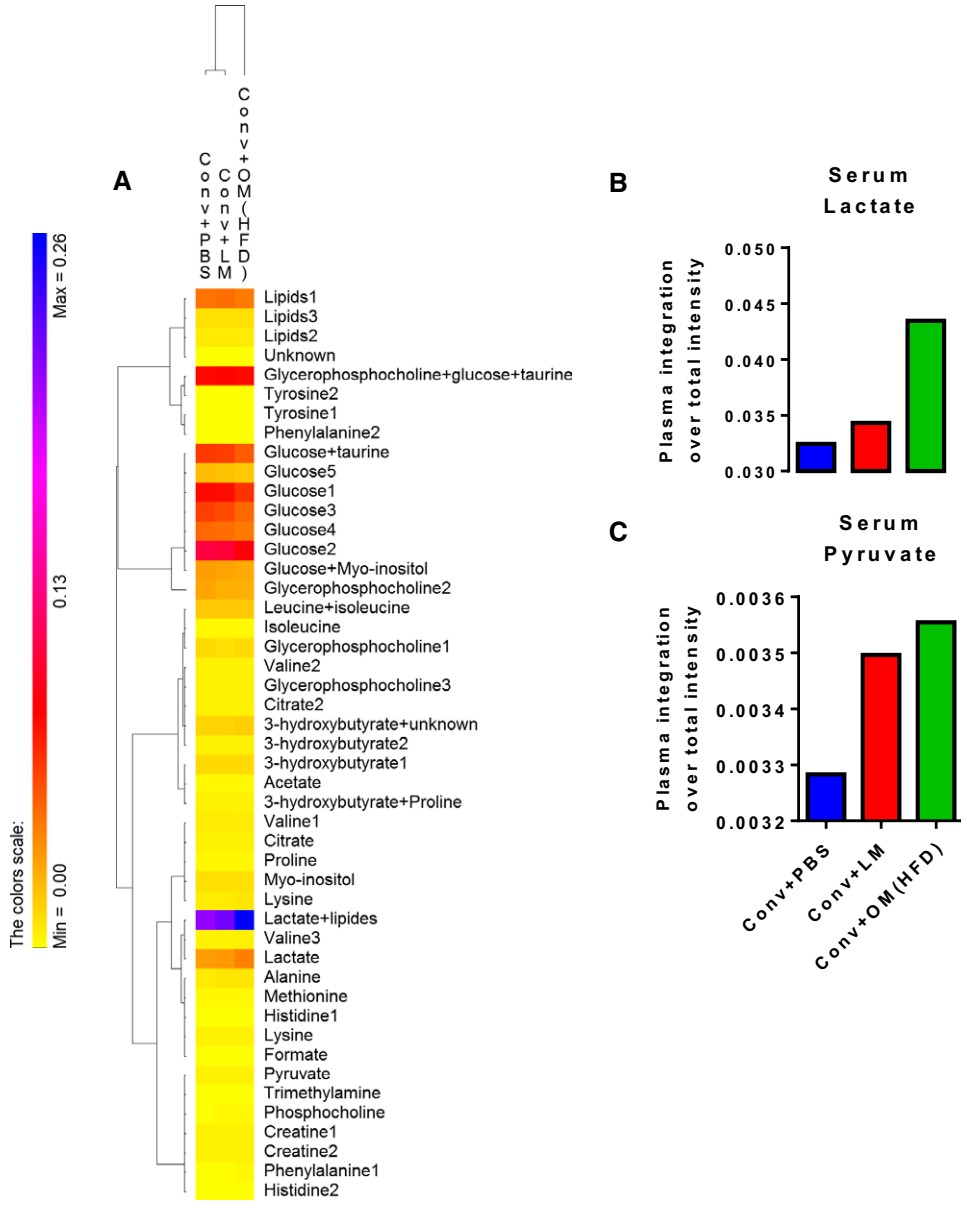

**Figure 2.  Transfer of dysbiotic vs. eubiotic gut microbiota in NC-fed conventional mice affects serum metabolome.**

A–C    (A) Heat-map analysis of serum metabolome; detailed histograms for (B) serum lactate and (C) serum pyruvate in antibiotic-free NC-fed conventional mice inoculated with either the vehicle (PBS) or caecal microbiota from either lean mice or HFD-fed mice (Conv + PBS, Conv + LM, Conv + OM(HFD), respectively). A pool of serum samples was used per group (*n* = 6).

expression in mice inoculated with the HFD-microbiota; only mice receiving the lean microbiota displayed a significant increase in IL-17a expression (Fig 3F). These observations are most likely due to the higher bacterial amount of the transplant from lean mice compared to the one from obese mice (Appendix Fig S1F–I). These data were in accordance with unchanged defensins production of both inoculated groups of mice (Fig 3G). Overall, none of the transfer significantly affected the general architecture of the ileum (Fig 3H).

These data show that the transfer of neither HFD- nor lean microbiota plays a major role in the modulation of intestinal, systemic and hepatic inflammation, excluding the involvement of these processes in the modulation of hepatic gluconeogenesis.

## Analysis of gut microbiota and microbiome in conventional mice fed a NC and inoculated with either a dysbiotic or eubiotic gut microbiota

In the light of the hepatic phenotype observed above and the role of gut microbiota dysbiosis on the liver (Dumas *et al*, 2006; Le Roy *et al*, 2013), we investigated the putative changes of the gut microbiota of recipient mice induced by the transfer.

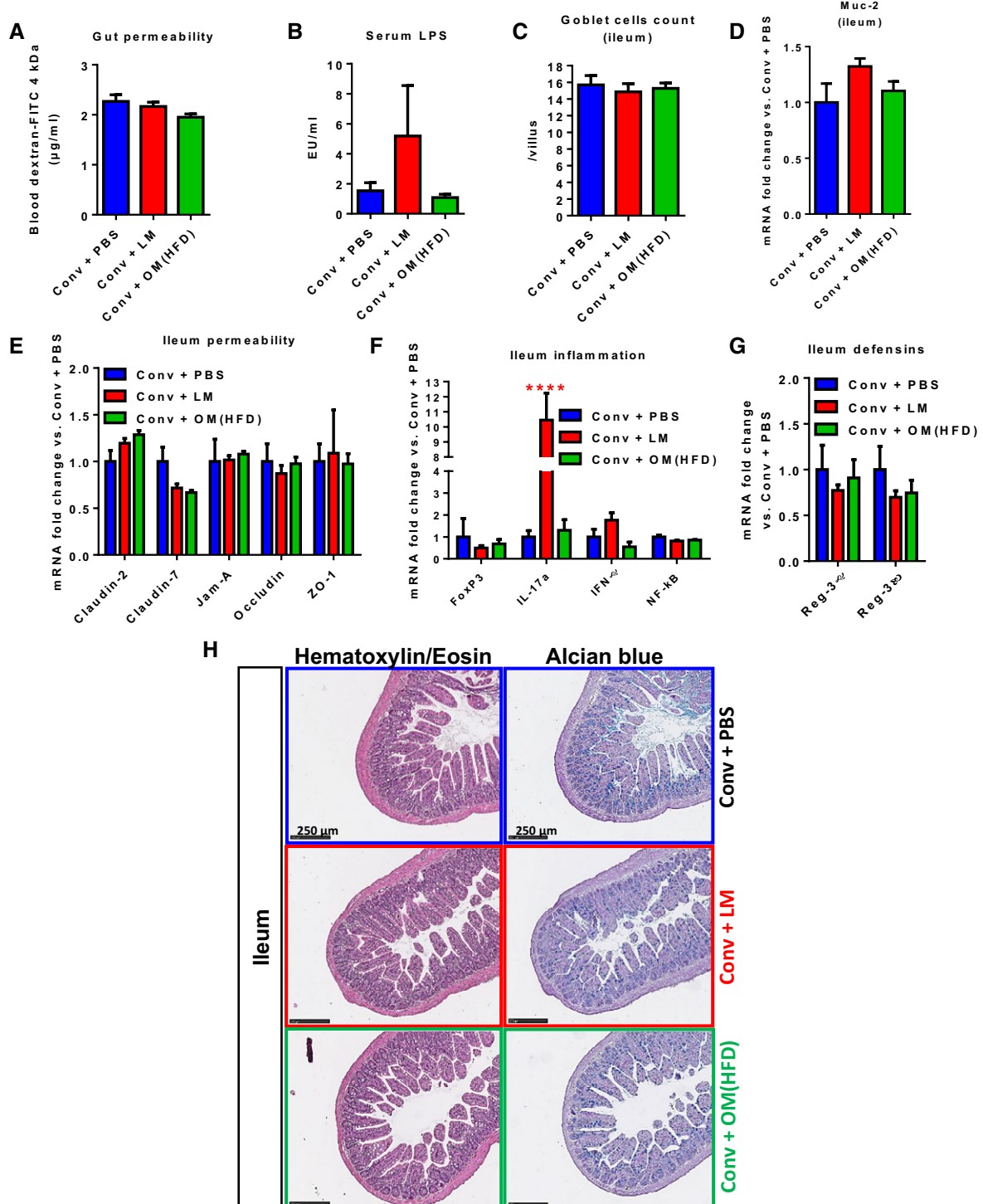

**Figure 3. Intestinal impact of transfer of dysbiotic vs. eubiotic gut microbiota in NC-fed conventional mice.**

A–H  (A) *In vivo* gut permeability; (B) serum LPS; (C) ileum goblet cell count/villus; ileum gene expression for (D) Muc-2, (E) tight junction proteins, (F) inflammatory markers, (G) defensins and (H) ileum histology in antibiotic-free NC-fed conventional mice inoculated with either the vehicle (PBS) or caecal microbiota from either lean mice or HFD-fed mice (Conv + PBS, Conv + LM, Conv + OM(HFD), respectively). Data are shown as mean ± SEM; *n* = 6, ****P < 0.0001; two-way ANOVA and Dunnett's post-test *vs.* Conv + PBS (F).

We analysed faeces microbiota at both taxonomic and related functional levels before (Basal) and after the transfer (Trans NC).

At baseline (Basal), the microbiota of the different groups displayed a certain degree of divergence, especially for the group to be inoculated with the HFD-microbiota, as shown by principal

coordinate analysis (PCoA) (Fig 4A, upper panel). In fact, mice designated to belong to the control group showed higher amount of *Parabacteroides* order; mice to be inoculated with the eubiotic microbiota showed higher amount of *Firmicutes* and mice to be inoculated with the dysbiotic microbiota had a higher amount of *Bacteroidetes* (Fig 4A, lower panel). Crucially, the metagenomic changes observed at baseline did not impact on basal hepatic

glucose production (Appendix Fig S1C). Thus, it is likely that the reduced hepatic gluconeogenesis can be ascribed to the transfer of gut microbiota.

After the inoculation, the three gut microbiota profiles presented some overlap (Fig 4B, upper panel). However, *Actinobacteria* taxon was significantly higher in mice receiving the eubiotic microbiota, whereas mice inoculated with dysbiotic microbiota showed higher

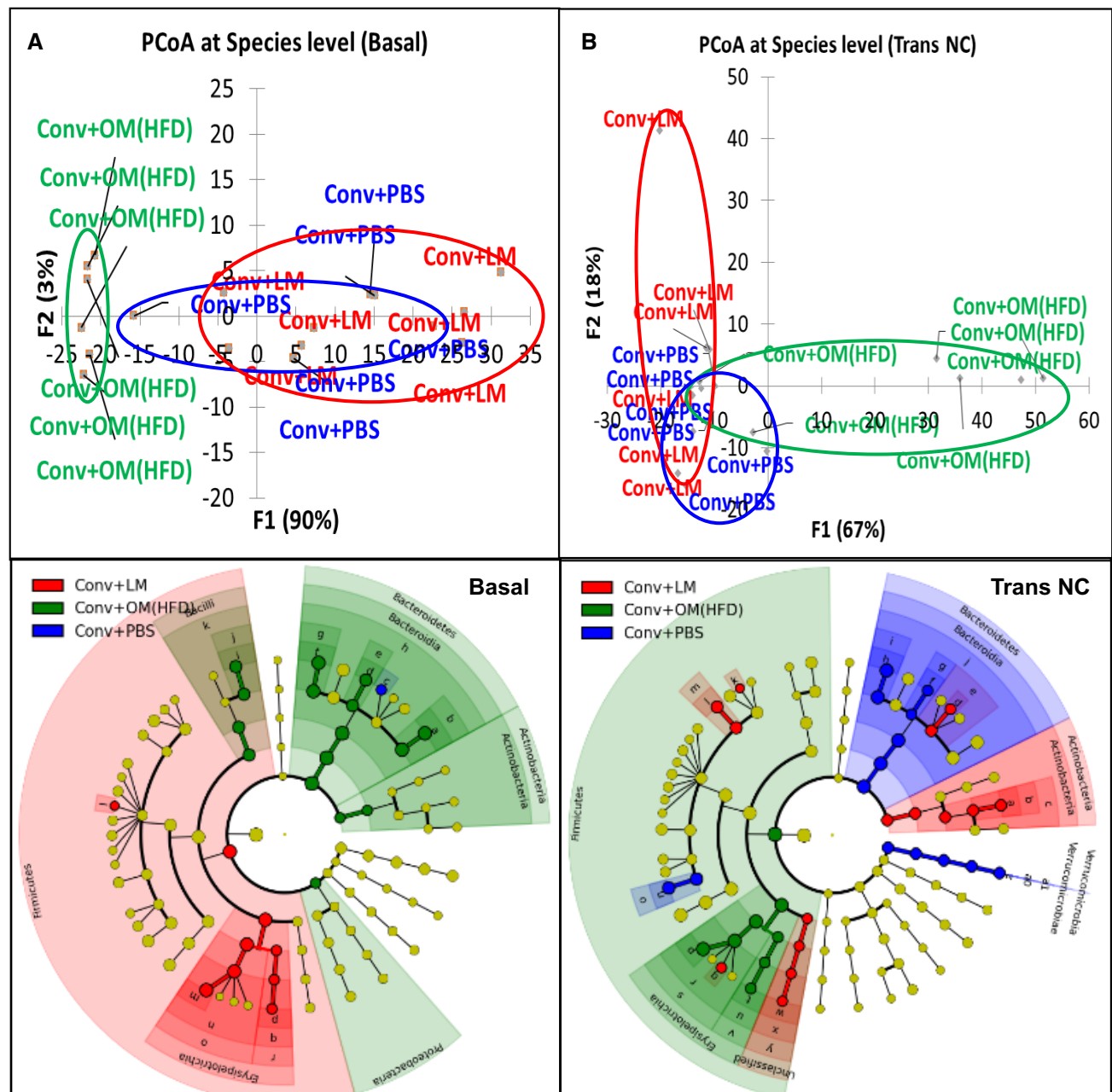

**Figure 4.  Effects of transfer of dysbiotic and eubiotic gut microbiota to NC-fed conventional mice on gut microbiota.**

A, B   (A) Principal coordinate analysis (PCoA) for gut microbiota at baseline (Basal) (upper panel) and related cladogram showing bacterial taxa significantly enriched in each group (lower panel); (B) PCoA for gut microbiota after transfer on NC (Trans NC) (upper panel) and related cladogram (lower panel) in antibiotic-free NC-fed conventional mice inoculated with either the vehicle or caecal microbiota from lean mice or HFD-fed mice (Conv + PBS, Conv + LM, Conv + OM(HFD), respectively) (n = 6).

amount of *Firmicutes* (Fig 4B, lower panel; The full list for clado-grams in Fig 4 is reported in Appendix Fig S4).

With regard to the microbiome analysed by PICRUSt (Langille *et al*, 2013), at baseline (Basal), mice displayed some functional divergences (Fig 5A–C), in accordance with gut microbiota profiles reported above (Fig 4A). After the inoculation, the dysbiotic micro-biota changed the gut microbiome of recipient mice differently from what did the eubiotic microbiota, as reported by the PCA analysis compared to control mice (Fig 5D). In fact, mice inoculated with either HFD-microbiota or lean microbiota did not share any micro-bial pathway (Fig 5E and F).

To evaluate the net impact of microbial transfer on the metage-nomic profile of recipient mice, we performed an *intragroup analy-sis* for both gut microbiota and microbiome. The gut microbiota of control mice showed some divergences compared to baseline (Appendix Fig S5A). In inoculated mice, the modifications induced by the lean microbiota were similar to the ones induced by PBS, except for the modulation of phylum *Tenericutes* (Appendix Fig S5B, lower panel). By contrast, the modifications induced by the HFD-microbiota were more distinct from the two other groups (Appendix Fig S5C, lower panel). This evidence suggests that the transfer of HFD-microbiota changed the gut microbiota of recipient mice to a greater extent than lean microbiota. This datum is in strong accordance with the higher metabolic modulation induced by the HFD-microbiota throughout the study.

The intragroup microbiome analysis reflected the aforementioned changes in gut microbiota, with the HFD-microbiota transfer showing the greater impact (Appendix Fig S5D–I); the control group and mice inoculated with the lean microbiota showed each some overlap (Appendix Fig S5D and F) with only three microbial pathways signifi-cantly affected (Appendix Fig S5E and G). By contrast, mice inoculated with the HFD-microbiota showed a distinct separation (Appendix Fig S5H), suggesting the greater impact of HFD-microbiota when compared to the lean microbiota. This result is also sustained by the greater number (twenty) of microbial pathways significantly modu-lated by the HFD-microbiota (Appendix Fig S5I) vs. the three microbial pathways found modulated above (Appendix Fig S5E and G).

These data show that the transfer of HFD-microbiota in antibi-otic-free NC-fed conventional mice is able to influence both micro-biota (taxonomy) and microbiome (function), to a greater extent than lean microbiota.

### Transfer of two different dysbiotic gut microbiota in conventional mice reduces markers of hepatic gluconeogenesis on NC and prevents hepatic alteration and adiposity on 72% HFD

To understand whether the origin of the dysbiosis (i.e. nutritional vs. genetic) could play a role in the observed metabolic modulation, we inoculated another group of mice (Conv + OM(ob)) with the gut microbiota from genetically obese mice (*ob/ob*, ob-microbiota here-after). On NC (Fig 6A), markers of hepatic gluconeogenesis were again lower in mice inoculated with the ob-microbiota compared to control mice, although not to the same extent as observed in mice receiving the HFD-microbiota (Appendix Fig S6A). No significant change was observed for body weight whatever the group (Appendix Fig S6C and D).

A subgroup of inoculated mice was then kept on NC (reported as "Normal chow long term"), and another subgroup was fed a

72% HFD (Fig 6A). The choice of this particular diet (Branchereau *et al*, 2016; Blasco-Baque *et al*, 2017) was based on its very low level of carbohydrates (< 1%). Therefore, in this model, glycaemia reflects hepatic gluconeogenesis. In the group of mice kept on NC, the reduced hepatic gluconeogenesis induced by inoculation was no longer observed (Appendix Fig S6N), suggest-ing an acute metabolic impact.

On 72% HFD, fed glycaemia was lower in mice inoculated with HFD-microbiota compared to control mice (Fig 6B). This datum was not associated with a change in fed insulinaemia (Appendix Fig S6B). Moreover, mice inoculated with the HFD-microbiota again showed a lower fasting glycaemia and a lower hepatic gluconeogenesis compared to control mice. Note also that a lower hepatic gluconeogenesis, but not a significant lower fasting glycaemia, was observed in 72% HFD-fed mice inoc-ulated with the ob-microbiota (Fig 6C). Again, we did not observe significant changes in PKA substrates phosphorylation, whereas we found a significant reduction in both amount and activity of PEPCK and G6Pase (Fig 6D and G). Again, the modulation of the activity of these key gluconeogenic enzymes together with a change in their protein amount provide a mechanism to explain and corroborate the observed modulation of hepatic glucose production, excluding the mere impact of reduced fasting glycaemia. Neither hepatic architecture nor liver weight, hepatic triglycerides and transaminases plasma levels were significantly affected (Appendix Fig S6E–I). By contrast, on NC, we observed a small improvement in glucose tolerance and a significant improvement in insulin tolerance (Appendix Fig S6J and K). The improved glucose tolerance, but not insulin tolerance, was kept on 72% HFD (Appendix Fig S6L and M).

We recently showed that the microbial sensor NOD2 mediates the onset of metabolic diseases in mice (Denou *et al*, 2015); there-fore, to assess whether this receptor may be involved in the regula-tion of the observed hepatic phenotype, we inoculated conventional NOD2 KO mice. The lack of NOD2 microbial sensor blunted the reduction of hepatic gluconeogenesis induced in WT mice by the transfer with both HFD- and ob-microbiota (Appendix Fig S6O).

Since an earlier study had reported increased adiposity in 72% HFD-fed mice (Serino *et al*, 2012b), we analysed the effect of gut microbiota transfer in white adipose tissue (WAT). Despite a lack of significant change in body weight, fat and lean mass (Appendix Fig S7A–C), mice inoculated with the HFD-microbiota displayed significant smaller adipocytes (Appendix Fig S7D) compared to control mice. Furthermore, these mice showed signif-icantly higher free fatty acids (FFA) plasma levels (Appendix Fig S7E) compared to control mice, whereas inoculation did not significantly affect plasma levels of total cholesterol, triglycerides, HDL and LDL lipoproteins (Appendix Fig S7F–I). By contrast, ob-microbiota had no significant impact on these parameters (Appendix Fig S7A–I). This result suggests that a genetic vs. nutri-tional dysbiosis of gut microbiota may have a divergent metabolic impact on WAT.

We also investigated whether the different origins of dysbiotic gut microbiota may affect intestinal inflammation and permeability. Inoculation with HFD-microbiota induced a significant increase in the ileum of iNOS, IFNγ and IL-6 gene expression and a tendency to increase the majority of the analysed inflammatory markers, also shown in mesenteric lymph nodes, whereas the ob-microbiota did

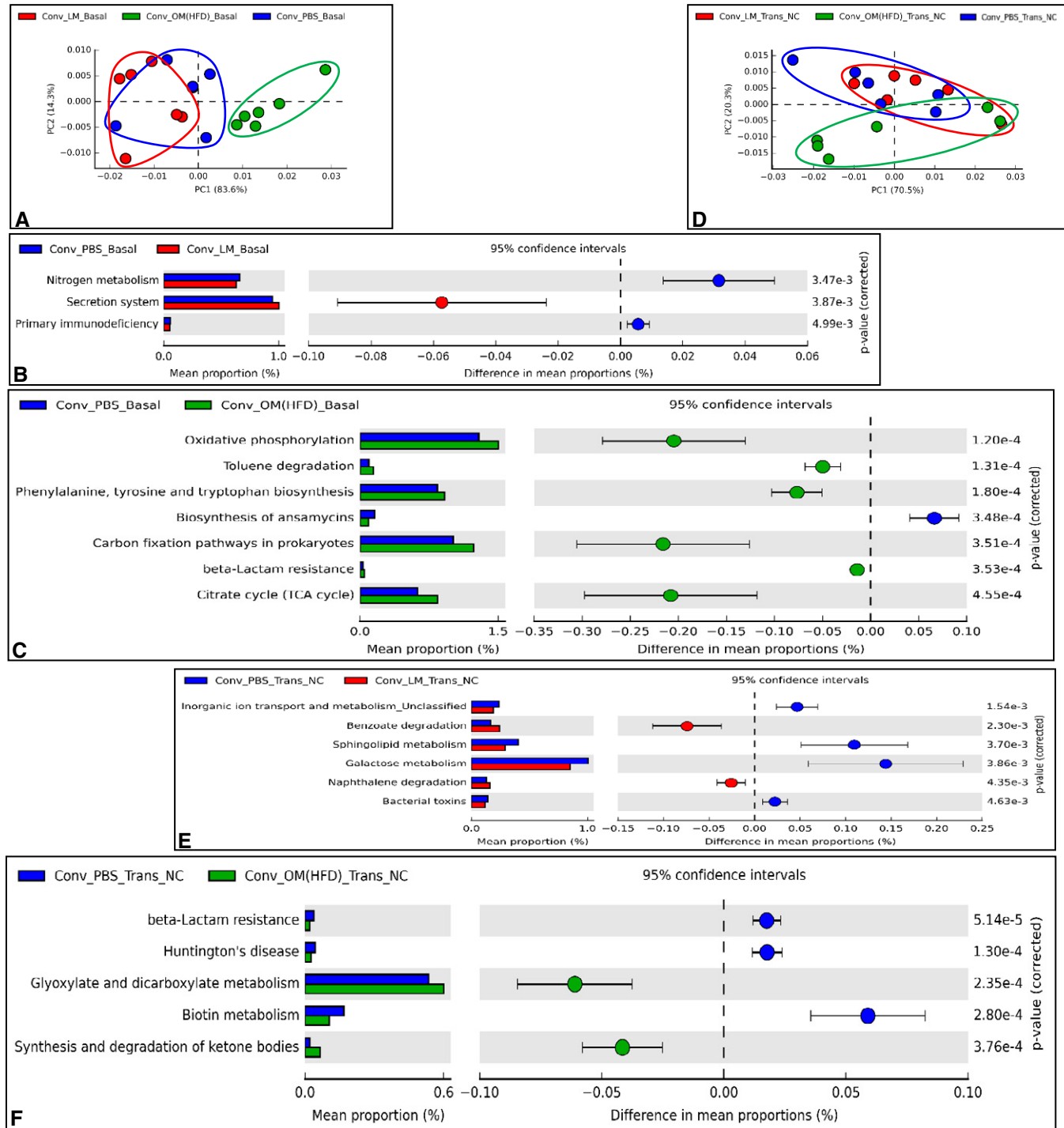

**Figure 5. Effects of transfer of dysbiotic and eubiotic gut microbiota to NC-fed conventional mice on gut microbiome.**

A–F  Principal component analysis showing PICRUSt-based gut microbiome study at baseline (Basal) (A) and after transfer on NC (Trans NC) (D) and top modulated (based on the two-sided Welch's *t*-test) microbial pathways in a pair-wise comparison (B, C, E, F) in antibiotic-free NC-fed conventional mice inoculated with either the vehicle or caecal microbiota from lean mice or HFD-fed mice (Conv + PBS, Conv + LM, Conv + OM(HFD), respectively) ($n = 6$).

not significantly affect these parameters (Appendix Fig S8A and B). Neither dysbiotic gut microbiota significantly changed the expression of tight junction proteins, whereas both transfers induced a tendency to increase defensins production (Appendix Fig S8C and D).

Altogether, these data show that mice inoculated with either HFD- or ob-microbiota had lower hepatic gluconeogenesis both following acute NC diet and 72% HFD, smaller WAT cell size and a minor intestinal inflammation with no change in intestinal permeability.

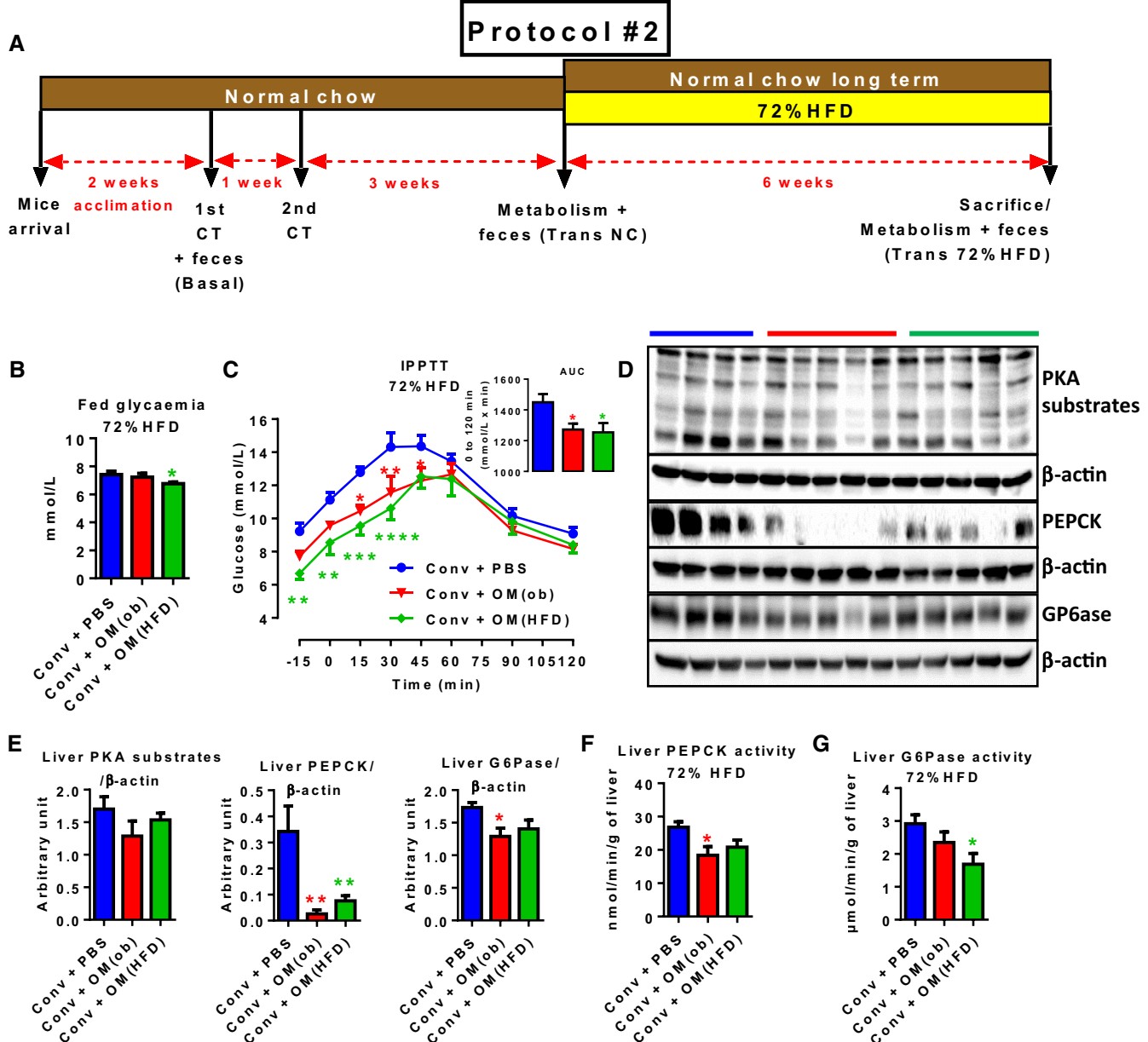

**Figure 6.  Transfer of dysbiotic gut microbiota in conventional mice prevents HFD-increased hepatic gluconeogenesis.**

A–G   (A) Experimental timeline: $1^{st}/2^{nd}$ CT (caecal transfer); after switching on 72% HFD: (B) fed glycaemia; (C) intraperitoneal pyruvate tolerance test and AUC as inset; (D) liver Western blot analyses for PKA substrates phosphorylation, PEPCK and G6Pase, all normalized on β-actin (loading control, an individual mouse per lane is shown) and related histograms (E); hepatic (F) PEPCK and (G) G6Pase enzymatic activity in antibiotic-free NC-fed conventional mice inoculated with either the vehicle or caecal microbiota from C57Bl/6 *ob/ob* or HFD-fed mice (Conv + PBS, + OM(ob), + OM(HFD), respectively). Data are shown as mean ± SEM; *n* = 5–6, \*$P < 0.05$, \*\*$P < 0.01$, \*\*\*$P < 0.001$, \*\*\*\*$P < 0.0001$; unpaired Student's *t*-test vs. Conv + PBS (for B, C inset, E–G) and two-way ANOVA with Dunnett's post-test vs. Conv + PBS (C).

Source data are available online for this figure.

## Analysis of gut microbiota and microbiome in conventional mice inoculated with a dysbiotic gut microbiota and fed a NC and a 72% HFD

We analysed both taxa and related metabolic functions three times: before inoculation (Basal); after the inoculation while the recipient mice were on NC (Trans NC) and after the inoculated mice were fed a 72% HFD (Trans 72% HFD).

At baseline (Basal), mice displayed again some divergences in microbial taxonomy, as reported by PCoA (Fig 7A, upper panel). Indeed, mice designated to belong to the control group (blue) presented a higher amount of *Bacteroides* and *Clostridium* genera; mice designated to receive the ob-microbiota (green) had a higher amount of *Proteobacteria* and *Bacteroidetes*; mice designated to receive the HFD-microbiota (red) had a higher amount of *Firmicutes* (Fig 7A, lower panel). Once more, the metagenomic differences

observed at baseline did not affect basal hepatic glucose production (Appendix Fig S1C).

After the transfer, mice inoculated with the HFD-microbiota and fed a NC showed a gut microbiota deeply different than control mice (Fig 7B, in green in the upper panel and red in the lower panel). By contrast, the gut microbiota of mice inoculated with the ob-microbiota was almost similar to the one of control mice (Fig 7B, in red in the upper panel and green in the lower panel). However, mice inoculated with the ob-microbiota had higher *Actinobacteria*, whereas control mice showed higher *Bacilli* and *Bacteroidia* classes (Fig 7B, lower panel).

When inoculated mice were fed a 72% HFD, the gut microbiota of both inoculated groups appeared more similar to the one of control group (Fig 7C, upper panel). This datum is in accordance with the strong ability of diet to affect gut microbiota (Carmody *et al*, 2015). Nevertheless, some bacterial taxa were still significantly different in each group of inoculated mice (Fig 7C, lower panel; The full list for cladograms in Fig 7 is reported in Appendix Fig S9).

With regard to the microbiome, at baseline (Basal), mice displayed a high degree of overlap, except for two mice (Fig 8A), although a few microbial pathways were found significantly modulated (Fig 8B and C). After the transfer, on NC, the separation of the three gut microbiome profiles (Fig 8D) was similar to the separation of gut microbiota profiles (Fig 7B). Note that microbial pathways significantly modulated compared to control mice were identified only in mice inoculated with the HFD-microbiota (Fig 8E).

We observed a change in markers of hepatic gluconeogenesis in both protocols (#1 and #2), which prompted us to look for a common microbial pathway associated to this hepatic phenotype by comparing the two microbiome analyses. The glyoxylate and dicarboxylate microbial pathway was the only found to be significantly modulated in both protocols and increased by the HFD-microbiota (Figs 5F and 8E). This microbial pathway showed a strong negative and significant correlation with the IPPTT area under the curve (AUC) (Appendix Fig S10), suggesting a putative role for glyoxylate and dicarboxylate microbial metabolism in the modulation of hepatic glucose production.

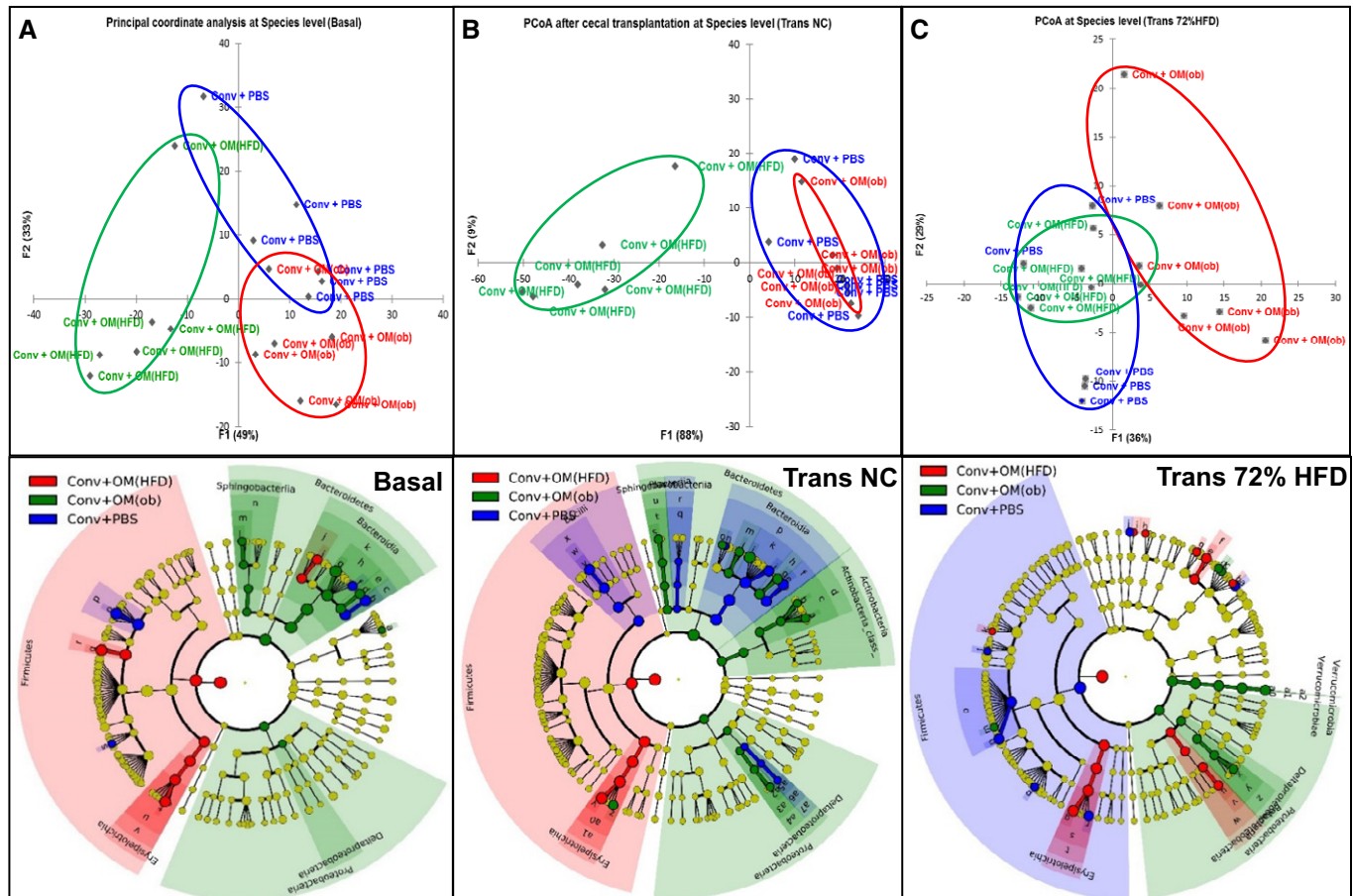

**Figure 7.  Transfer of dysbiotic gut microbiota in conventional mice changes gut microbiota according to the origin of dysbiosis on both NC and 72% HFD.**

A–C   (A) Principal coordinate analysis (PCoA) for gut microbiota at baseline (Basal) (upper panel) and related cladogram showing bacterial taxa significantly enriched in each group (lower panel); (B) PCoA for gut microbiota after transfer on NC (Trans NC) (upper panel) and related cladogram (lower panel); (C) PCoA for gut microbiota after transfer on 72% HFD (Trans 72% HFD) (upper panel) and related cladogram (lower panel) in antibiotic-free NC-fed conventional mice inoculated with either the vehicle or caecal microbiota from C57Bl/6 *ob/ob* or HFD-fed mice (Conv + PBS, + OM(ob), + OM(HFD), respectively) and then fed a 72% HFD (*n* = 5–6).

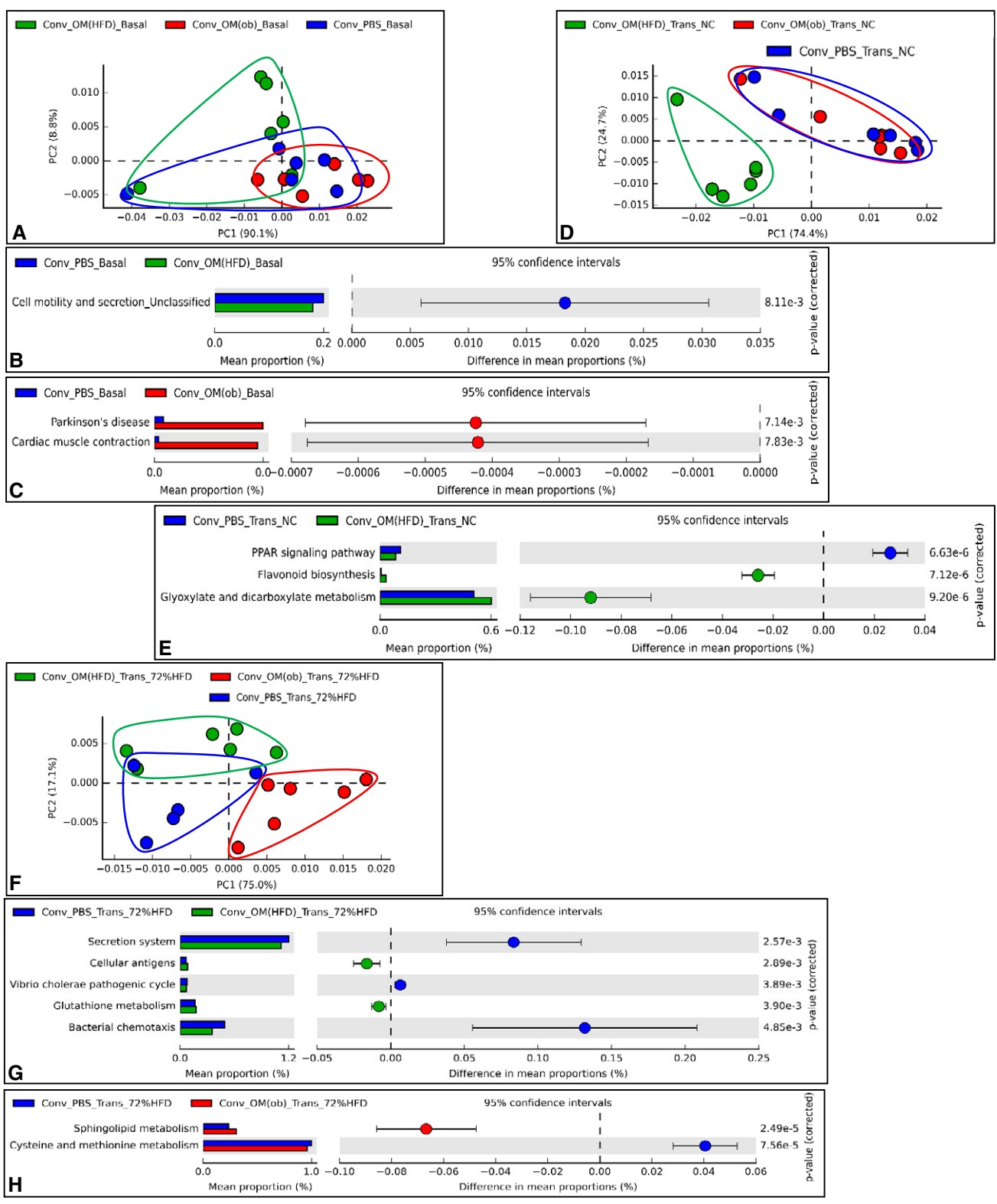

**Figure 8.   Transfer of dysbiotic gut microbiota in conventional mice changes gut microbiota and microbiome according to the origin of dysbiosis on both NC and 72% HFD.**

A–H    Principal component analysis showing PICRUSt-based gut microbiome study at baseline (Basal) (A), after transfer on NC (Trans NC) (D) and on 72% HFD (Trans 72% HFD) (F) and top modulated (based on the two-sided Welch's *t*-test) microbial pathways in a pair-wise comparison (B, C, E, G, H) in antibiotic-free NC-fed conventional mice inoculated with either the vehicle or caecal microbiota from C57Bl/6 *ob/ob* or HFD-fed mice (Conv + PBS, + OM(ob), + OM(HFD), respectively) and then fed a 72% HFD (*n* = 5–6).

With regard to the microbiome analysis of inoculated mice once fed a 72% HFD, mice inoculated with the ob-microbiota showed some divergences compared to control mice (Fig 8F). However, in both groups of inoculated mice, we could identify some significantly modulated microbial pathways (Fig 8G and H).

To evaluate the net impact of the transfer on the metagenomic profile of recipient mice, we performed an *intragroup analysis* for both gut microbiota and microbiome. The gut microbiota of control mice showed some differences compared to baseline (Appendix Fig S11A). Both transfers changed the gut microbiota of recipient mice (Appendix Fig S11B and C) as also observed for the microbiome (Appendix Fig S11D–I). Moreover, the HFD-microbiota affected a greater number (ten) of microbial pathways (Appendix Fig S11I) compared to the ob-microbiota (one) (Appendix Fig S11G). We also performed an *intragroup analysis* for both gut microbiota and microbiome comparing the NC and 72% HFD nutritional states. With regard to the gut microbiota, this analysis showed that the impact of 72% HFD was important in all groups, but with a greater extent in mice inoculated with the HFD-microbiota (Appendix Fig S12A–C). In terms of gut microbiome, the microbial pathways modulated were highly specific to each group. The glyoxylate and dicarboxylate microbial pathway was increased only in HFD-microbiota inoculated mice when fed a NC. This suggests that this pathway would not totally account for the gluconeogenesis reduction still observed once inoculated mice were fed a 72% HFD.

These data show the divergent impact of the two dysbiotic gut microbiota on both gut microbiota and microbiome on NC and 72% HFD.

## Discussion

In this study, we report that antibiotic-free conventional mice inoculated with a dysbiotic gut microbiota from either HFD-induced or *ob/ob* obese mice unexpectedly show acute lower hepatic gluconeogenesis on NC and protection from 72% HFD-increased hepatic gluconeogenesis and adiposity. These phenotypic traits were associated with changes in both gut microbiota and microbiome. Mice inoculated with HFD-microbiota showed in both protocols reduced (i) fasting glycaemia and (ii) markers of hepatic gluconeogenesis, and higher (iii) *Firmicutes* and (iv) glyoxylate and dicarboxylate microbial pathway on NC. By contrast, transferring gut microbiota from lean mice did not affect hepatic metabolism, despite some changes in both gut microbiota and microbiome. Indeed, the modulation of markers of hepatic gluconeogenesis was in accordance with the decreased amount and/or activity of key gluconeogenic enzymes, depending on the diet of recipient mice. The 72% HFD may account for discrepancies observed for PEPCK regulation during the dysbiotic transfer, mostly by affecting the gut microbiota of the recipient, leading to the consequent systemic effects.

Mice inoculated with HFD-microbiota showed higher plasma levels of lactate and pyruvate, in accordance with reduced hepatic glucose production (Madiraju *et al*, 2014). This reduction was not due to hepatic damage, since the liver of inoculated mice showed no inflammation nor increased transaminases and triglycerides. This hepatic phenotype was blunted in NOD2 KO mice, suggesting the

involvement of NOD2 microbial sensor in the management of the metabolic effects induced by the inoculation of gut microbiota in recipient mice. In both protocols used, mice inoculated with HFD-microbiota showed a higher relative abundance of *Firmicutes*, Gram-positive bacteria harbouring a more developed peptidoglycan than Gram-negative ones. Given that peptidoglycan is a NOD2 ligand, we may speculate that NOD2 activation may be implicated in the observed hepatic phenotype. The diverse gut microbiota harboured by NOD2 KO mice (Mondot *et al*, 2012; Denou *et al*, 2015) may limit the aforementioned reduction in hepatic glucose production induced by the inoculation.

In general, we observed contrasting results compared to the metabolic impact of eubiotic gut microbiota transfer in metabolic syndrome patients (Vrieze *et al*, 2012) and of dysbiotic gut microbiota in axenic mice (Turnbaugh *et al*, 2006). The explanation we propose is that the functional gut barrier and mature immune system of a conventional mouse may allow a better management of dysbiotic gut microbiota than in axenic mice. This may result in a more efficient immune response, as suggested by increased IL-17a gene expression in the intestine of mice inoculated with lean microbiota. By contrast, HFD-microbiota did not induce this raise, in accordance with our previous report (Garidou *et al*, 2015). Moreover, gut barrier impairment is essential for dysbiosis-induced metabolic alterations (Serino *et al*, 2014). Hence, hyper-permeability and altered villi architecture typifying axenic intestine (Reinhardt *et al*, 2012) may favour an uncontrolled spread of both bacteria and their antigens. This may trigger a metabolic inflammation (Amar *et al*, 2011) responsible for dysbiosis-induced metabolic alterations reported in axenic mice. This rationale is sustained by our results showing that the gut barrier is not affected by transferring the gut microbiota into a conventional mouse.

The efficient immune response may have systemic beneficial impacts such as the ones observed on the liver and the WAT. With regard to the latter, in mice inoculated with the HFD-microbiota the reduced expression of *de novo* lipogenic genes is in accordance with smaller adipocytes. In fact, Eissing *et al* (2013) showed that lipogenic enzymes are upregulated in the liver of obese patients, and in our study, lipogenic enzymes are downregulated in the liver of mice in association with smaller adipocytes. Increased serum FFA were associated with smaller adipocytes in axenic mice (Backhed *et al*, 2004) too, in accordance with our data.

Overall, to explain our counterintuitive results, we analysed the inflammation of recipient mice. We note that the inoculation of dysbiotic gut microbiota in conventional mice induced the up-regulation of the effector response (IFNγ) and the down-regulation of the regulatory response (FoxP3), in the ileum and mesenteric lymph node, similarly to axenic mice colonized with normal gut microbiota (Naik *et al*, 2012). These results may be dependent on the capacity of a conventional mouse to develop an effective response towards the "obese" antigens due to a mature immune system and a functional gut barrier. The significant negative correlation between the glyoxylate and dicarboxylate microbial pathway and the IPPTT AUC suggests a link between this microbial activity and the regulation of hepatic glucose production. Our hypothesis is supported by a recent publication showing that the glyoxylate and dicarboxylate microbial pathway is among the most affected in the model of Zucker diabetic fatty rats (Dong *et al*, 2016). Therefore, targeting microbial genes involved in this pathway may

be effective for the control of hepatic function on a NC feeding, but no longer on 72% HFD.

In conclusion, our results could open a new debate on the impact of gut microbiota dysbiosis on host metabolism by describing the beneficial effects of the transfer of dysbiotic gut microbiota, principally on the liver. Thus, our new observation may encourage re-examining the causal role of gut microbiota dysbiosis on metabolic diseases.

# Materials and Methods

### Animal model and diet

Six-week-old C57Bl/6 (WT or NOD2 KO) male mice (Charles River, L'Arbresle, France) were fed a normal chow (NC) for 4 weeks (protocol #1) or a NC and then a high-fat diet (HFD) (~72% fat (corn-oil and lard), 28% protein and < 1% carbohydrate; SAFE, Augy, France) (Serino *et al*, 2012b) for 6 weeks (protocol #2). Mice were group-housed (5 or 6 mice per cage) in a specific-pathogen-free controlled environment (inverted 12-h daylight cycle, light off at 10:00 a.m.). Six-hour-fasted mice were sacrificed by cervical dislocation. Then, tissues were collected and snap-frozen in liquid nitrogen. All animal experimental procedures were approved by the local ethical committee of Rangueil University Hospital (Toulouse, France).

### Gut microbiota transfer

Two protocols were performed: recipient mice were NC-fed 6-week-old C57Bl/6 male mice (Charles River, L'Arbresle, France), inoculated in a fed condition and never treated previously with antibiotics for both protocols.

*Protocol #1*
Donor mice: Eight-week-old C57Bl/6 male mice (Charles River, L'Arbresle, France) were either fed a 60% HFD (60% fat, 20% carbohydrates, 20% proteins) (Serino *et al*, 2007) or a NC for 3 months. Then, the caecum content from these mice served as transplant and was suspended in sterile reduced PBS (N$_2$ gas and thioglycolic acid, Sigma Aldrich, St. Louis, MO) at the concentration of 200 mg/ml. Non-antibiotic treated 6-week-old conventional C57Bl/6 male mice (Charles River, L'Arbresle, France) served as recipient mice and were gavaged with 200 μl of either sterile reduced PBS (Conv + PBS) or 200 μl at 200 mg/ml caecum suspension of either eubiotic gut microbiota from lean mice (Conv + LM) or dysbiotic gut microbiota from HFD-induced obese mice (Conv + OM(HFD)) once per week, for 2 weeks ("LM" stands for lean microbiota and "OM" stands for obese microbiota). The caecum content from 3 to 6 mice per group of donors was pooled and provided to recipient mice at the same concentration of 200 mg/ml.

*Protocol #2*
Donor mice: Eleven- to twelve-week-old C57Bl/6 male *ob/ob* mice (Charles River, L'Arbresle, France) or 20-week-old C57Bl/6 male mice (Charles River, L'Arbresle, France) fed a 60% HFD (Serino *et al*, 2007) served as donor mice. Then, the caecum content of these mice served as transplant and was suspended in sterile reduced PBS (N$_2$ gas and thioglycolic acid, Sigma Aldrich, St. Louis,

MO). Non-antibiotic treated 6-week-old conventional C57Bl/6 male mice (Charles River, L'Arbresle, France) served as recipient mice and were gavaged with 200 μl of either sterile reduced PBS (Conv + PBS) or 200 μl at 200 mg/ml caecum suspension of dysbiotic gut microbiota from either *ob/ob* mice (Conv + OM(ob)) or from HFD-induced obese mice (Conv + OM(HFD)) once per week, for 2 weeks ("OM" stands for obese microbiota). The caecum content from 3 to 6 mice per group of donors was pooled and provided to recipient mice at the same concentration of 200 mg/ml. Note that the unmatched age for donor mice in protocol #2 is related to the fact that the major point we wanted to investigate herein is the putative metabolic effect of transferring a dysbiotic gut microbiota; therefore, we did not intend to compare donors of protocol #2 against each other.

*Criteria for the definition of eubiotic vs. dysbiotic gut microbiota*
Eubiotic vs. dysbiotic gut microbiota were defined according to the amount of bacteria, lower in the dysbiotic gut microbiota and their high diversity according to the donor (NC vs. HFD-fed mouse; Appendix Fig S1F–K).

### Western blot analysis

The Western blot analysis in liver extracts was performed as previously described (Serino *et al*, 2012b). The following antibodies were used: PKA substrates, β-actin, PEPCK, all from Cell Signalling Technology. The G6Pase antibody (De Vadder *et al*, 2014) was kindly provided by Dr. Gilles Mithieux and Dr. Fabienne Rajas (see Acknowledgements).

### Hepatic glycogen dosage

50–100 mg of liver from 6-h-fasted mice was dissolved in 200 μl of 1 M NaOH at 55°C for 1 h. Samples were neutralized with 200 μl 1 N HCl and then centrifuged at 7,000 *g* for 5 min at 4°C. Then, to hydrolyse the hepatic glycogen content, 10 μl of supernatant were incubated in 40 μl of a solution of 50 U/ml amyloglucosidase (Sigma) diluted in 0.2 M sodium acetate buffer at pH 7.4. As a control, 10 μl were incubated in 40 μl of sodium acetate buffer only. The tubes were incubated for 1 h at 55°C. Then, glucose concentration was measured with Glucose GOD FS reagent (DiaSys Diagnostic Systems GmbH) according to manufacturer's instructions. The difference of glucose concentration between the two conditions with and without amyloglucosidase represented the hepatic glycogen content per sample. Glycogen was expressed as micrograms of glucose resulting from glycogen hydrolysis per milligrams of liver.

### Intraperitoneal (IP) pyruvate tolerance test (IPPTT), glucose tolerance test (IPGTT) and insulin tolerance test (IPITT) or oral glucose (OGTT) tolerance test

Since mice were on an inverted light-cycle, IPPTT was performed by injecting pyruvate (2 g/kg) in 6-h-fasted mice (Ribeiro *et al*, 2016). Glycaemia was measured as previously described (Cani *et al*, 2007) at −15, 0, 15, 30, 45, 60, 90 and 120 min. OGTT was performed as described elsewhere (Cani *et al*, 2007). For IPITT, 3-h fasted mice were injected with 0.75 U/kg insulin (Serino *et al*, 2007). Area under the curve (AUC) is also shown as inset for IPPTTs and

IPGTTs/OGTT. AUC was calculated by the trapezoidal rule (Le Floch *et al*, 1990) using GraphPad Prism version 7.00 for Windows Vista (GraphPad Software, San Diego, CA) and shown as mmol/l × min.

## Liver triglycerides measurement

Liver triglycerides have been measured using the Free Glycerol Reagent and Triglyceride Reagent, both from Sigma (Sigma Aldrich, St. Louis, MO).

## Adipocyte size determination

Epididymal WAT was collected and fixed in 70% ethanol. The tissue was processed on the STP 120 Spin Tissue Processor by ethanol dehydration (increasing bath from 70% to 100%), xylene substitution and paraffin infiltration. 5-µm paraffin sections were obtained using a Microtome Microm HM 340E and stained by haematoxylin/eosin. Stained sections were imaged on a Zeiss PALM MicroBeam system with Plan-Neofluar 10× (0.3 NA) air objectives and AxioCam MRm black and white camera. Images were analysed using MotionTracking software (Collinet *et al*, 2010) following a pipeline developed by Dr Giovanni Marsico. First, the centre of the adipocytes was manually located. Then, adipocyte border was automatically segmented by a region growing algorithm based on the watershed transform. Then, the size of the adipocyte was plotted as cumulative distribution.

## Fat/lean mass measurement

Fat/lean mass (%) was measured via the EchoMRI-100 TM 3 in 1 system (EchoMRI LLC, Houston, TX, USA).

## Biochemical assays

Plasma aspartate (AST) and alanine (ALT) transaminases, total cholesterol, high-/low-density lipoprotein (HDL and LDL, respectively), triglycerides and free fatty acids (FFA) were measured by multiplex assays by the Phenotypage-ANEXPLO Platform (US06-CREFRE).

## Metabolomic analysis

Plasma samples (100 µl, out of a pool of $n = 6$ mice per group) were diluted with 600 µl of deuterium oxide ($D_2O$) and centrifuged at 5,000 $g$ for 10 min before they were placed in 5-mm NMR tubes. $^1$H NMR spectra were obtained on a Bruker DRX-600 Avance NMR spectrometer operating at 600.13 MHz for $^1$H resonance frequency using an inverse detection 5 mm $^1$H-$^{13}$C-$^{15}$N cryoprobe attached to a CryoPlatform (the preamplifier cooling unit). The $^1$H NMR spectra were acquired at 300 K using the Carr-Purcell-Meiboom_Gill (CPMG) spin-echo pulse sequence with pre-saturation, with a total spin-echo delay (2 nt) of 64 ms to attenuate broad signals from proteins and lipoproteins. A total of 128 transients were collected into 32 k data points using a spectral width of 12 ppm, a relaxation delay of 5 s and an acquisition time of 2.28 s. Prior to Fourier transformation, an exponential line broadening function of 0.3 Hz was applied to the FID. NMR spectra were phased and baseline corrected, and then, metabolites signals were integrated, and normalized to the total spectral area.

## Enzymatic activities

Hepatic glucose-6 phosphatase activity was determined as previously described (Rajas *et al*, 1999). Hepatic phosphoenolpyruvate carboxykinase activity was determined with the method of Pogson and Smith (Pogson & Smith, 1975).

## Microarray gene expression study and String analysis

Gene expression analysis was performed at the GeT-TRiX facility (GénoToul, Génopole Toulouse Midi-Pyrénées) using Agilent SurePrint G3 Mouse GE v2 8x60K microarrays (design ID 074809) following the manufacturer's instructions (Agilent Technologies, Santa Clara, California). For each of the six samples, Cyanine-3 (Cy3)-labelled cRNA was prepared from 200 ng of total RNA using the One-Color Quick Amp Labeling kit (Agilent) according to the manufacturer's instructions, followed by Agencourt RNAClean XP (Agencourt Bioscience Corporation, Beverly, Massachusetts). 600 ng of Cy3-labelled cRNA was hybridized on the microarray slides following the manufacturer's instructions. Immediately after washing, the slides were scanned on Agilent G2505C Microarray Scanner using Agilent Scan Control A.8.5.1 software and a fluorescence signal extracted using Agilent Feature Extraction software v10.10.1.1 with default parameters (grid 074809_D_F_20150624 and protocol GE1_1010_Sep10). Genes were considered differently expressed between Conv + OM(HFD) vs. Conv + PBS and between Conv + LM vs. Conv + PBS groups when $P < 0.05$. We also considered a logarithm of the fold change vs. Conv + PBS between −2.2 (for downstream regulation) and 2.5 (for upstream regulation).

### String-based microarray data analysis

The lists of hepatic gene differently expressed between Conv + OM (HFD) vs. Conv + PBS and between Conv + LM vs. Conv + PBS groups were mapped using the STRING database (http://string-db.org/). Each gene is represented by a node, and the thickness of lines between nodes illustrates the strength of interactions based on the literature and databases.

## RNA extraction and qPCR in liver, ileum and MLN

Total RNA was extracted from frozen tissues using the miRNeasy mini kit (Qiagen, Courtaboeuf, France). For mRNA, qPCR was performed as previously described (Serino *et al*, 2012b), except for ileum and mesenteric lymph node (MLN), where 500 ng of cDNAs was amplified using the ViiA7 system (Applied Biosystems). Results were expressed as $2^{-\Delta\Delta Ct}$ as already described (Serino *et al*, 2012b) and shown after normalization by the mean of the control values (Conv + PBS). The housekeeping gene used in this study is the Ribosomal Protein L19 (RPL19).

All the primers used in this study are listed in Appendix Table S1.

## Taxonomic analysis of gut microbiota by pyrosequencing

Following Protocol #1 or #2, faecal total DNA was extracted as previously described (Serino *et al*, 2012b). The whole 16S bacterial DNA V2 region was targeted by the 28F-519R primers and pyrosequenced by the 454 FLX Roche technologies at Research&Testing

Laboratory (http://www.researchandtesting.com/, Texas, USA). An average of 3,000 sequences was generated per sample.

A complete description of bioinformatic filters can be found at http://www.rtlgenomics.com/docs/Data_Analysis_Methodology.pdf. Upper panels of Figs 4A and B, and 7A–C were drawn by XLSTAT for Windows Excel; lower panel cladograms of Figs 4A and B, and 7A–C were drawn by the Huttenhower Galaxy web application (http://huttenhower.sph.harvard.edu/galaxy/) website via the LEfSe algorithm (Segata *et al*, 2011).

### Functional analysis of the gut microbiota via microbiome analysis

Functional analysis of gut microbiota was performed via PICRUSt (Langille *et al*, 2013). Principal component analyses and extended error bar analyses with 95% confidence interval for Figs 5 and 8 were drawn via Statistical Analysis of Metagenomic Profiles (STAMP) software (Parks *et al*, 2014).

### Statistical analysis

Results are presented as means ± SEM. Statistical analyses were performed by one-way or two-way ANOVA followed by Sidak's or Dunnett's post-test, as reported or by unpaired Student's *t*-test, using GraphPad Prism version 7.00 for Windows Vista (GraphPad Software, San Diego, CA). Significant values considered at $P < 0.05$ or as reported. Figs 5 and 8 were analysed by a two-sided Welch's *t*-test; upper panels of Appendix Figs S1J/S5A–C/S11A–C/S12A–C were drawn by XLSTAT for Windows Excel.

### Data availability

The microarray data from this publication have been deposited to the Gene Expression Omnibus (GEO) database https://www.ncbi.nlm.nih.gov/geo/ and assigned the identifier (accession GSE81318).

The metabolomics data from this publication are available as electronic version Dataset EV1.

The metagenomics data from this publication have been deposited to the ENA database http://www.ebi.ac.uk/ena and assigned the identifier PRJEB19465.

**Expanded View** for this article is available online.

### Acknowledgements
We thank the zootechnie-Rangueil INSERM/UPS US006 CREFRE and the Phenotypage-ANEXPLO Platform (US006-CREFRE, Toulouse, France) for biochemical assays; Cecile Canlet and the Platform MetaToul-AXIOM for the NMR analyses; Lucie Fontaine from the histology facility of I2MC (Toulouse, France); Dr Jean-Philippe Pradère from I2MC for helping with liver histology; Claire Naylies and Yannick Lippi for microarray fingerprints acquisition and data analysis carried out at GeT-TRiX Genopole Toulouse Midi-Pyrénées facility. We warmly thank Dr. Gilles Mithieux and Dr. Fabienne Rajas from Unité 1213 "Nutrition Diabète et Cerveau" Faculté de Médecine Laënnec for kindly providing us with the antibody against G6Pase. Dr Tanti and Dr Cormont are supported by the LABEX SIGNALIFE: Grant ANR-11-LABX-0028-01. F.C. was supported by an INSERM/Région Provence Alpes-Cote d'Azur doctoral fellowship and by a grant from the Société Francophone du Diabète; they also thank the C3M histology and light microscopy facilities (Nice, France) for adipose tissue analyses and Dr. Giovanni Marsico from Max Planck Institute (Dresden, GERMANY) for the development of bioinformatics tools to measure adipocyte size. We tenderly thank Lorette Gaffié and Dr. Robert Cameron for their invaluable help with English editing. This work was supported in part by grants from Agence Nationale pour la Recherche (ANR) to RB and collaborators (ANR-Bactimmunodia and LPS diagra); in part, by the European Commission's Seventh Framework programme under grant agreement No. 241913 (FLORINASH) to RB and by the Institut Benjamin Delessert to MS. PDC is a research associate at FRS-FNRS (Fonds de la Recherche Scientifique), Belgium. PDC was supported by grants from the FRS-FNRS, the FRFS-WELBIO under grant: WELBIO-CR-2012S-02R, Funds Baillet Latour (Grant for Medical Research 2015), an ERC Starting Grant in 2013 (European Research Council, Starting grant 336452-ENIGMO). FC was supported by a fellowship from INSERM/Région PACA/FEDER and by a grant from the Société Francophone du Diabète (SFD). The C3M group was supported by the French National Research Agency (ANR) through the "Investments for the Future" LABEX SIGNALIFE (ANR-11-LABX-0028-01 grant).

### Author contributions
SN performed and analysed experiments; VB-B, AF, JG, PK, AW, FC, AM performed experiments; RP, JSI performed bioinformatic analysis on PICRUSt and microarray data; FT interpreted data and revised the manuscript; PDC performed plasma LPS dosage; J-FT, RB read the manuscript; CK, MC analysed data; MS designed, performed experiments, analysed and interpreted data and wrote the manuscript. All authors have approved the final version to be published.

### Conflict of interest
The authors declare that they have no conflict of interest.

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
