## [Review Process File · Molecular Systems Biology]

Transfer of dysbiotic gut microbiota has beneficial effects on host liver metabolism

Simon Nicolas, Vincent Blasco-Baque, Audren Fournel, Jerome Gilleron, Pascale Klopp, Aurelie Waget, Franck Ceppo, Alysson Marlin, Roshan Padmanabhan, Jason Iacovoni, François Tercé, Patrice Cani, Jean-Francois Tanti, Remy Burcelin, Claude Knauf, Mireille Cormont and Matteo Serino

Corresponding author: Matteo Serino, Digestive Health Research Institute (IRSD)

Review timeline:

Submission date:	28 September 2016
Editorial Decision:	21 November 2016
Revision received:	28 November 2016
Editorial Decision:	12 December 2016
Revision received:	25 January 2017
Editorial Decision:	08 February 2017
Revision received:	15 February 2017
Accepted:	20 February 2017

Editor: Maria Polychronidou

Transaction Report:

1st Editorial Decision

21 November 2016

We have now heard back from the two referees who agreed to evaluate your study. As you will see below, the reviewers raise a number of issues, which we would ask you to address in a revision. The referees' comments are quite clear so I think that there is no need to repeat the points listed below. Please let me know in case you would like to further discuss any specific point.

REFeree REPORTS

Reviewer #1:

In this manuscript, Serino and co-workers investigate an intriguing interaction between gut microbiota transfer and diet in glucose homeostasis, through a combination of physiological phenotyping, 16s rDNA pyrosequencing, metabolomics and transcriptomics. Using an antibiotics-free microbiota transfer approach, the authors obtain surprising results compared to previously reported studies. The authors observe a lower bacterial DNA load in the HFD inoculum, mostly related to a decrease of anaerobic bacteria. They also observe an increase in IL17 expression in the recipients of lean microbiota and a lower hepatic gluconeogenesis in the recipient of an obese microbiota fed a HFD, compared to recipients of lean microbiota or vehicle.

I would be grateful if the authors could take the following comments into account:

- The manuscript is generally well written and reports a wealth of data. The discussion should be tighter and focus only on the main findings to enhance clarity.
- Along the same lines the authors should make an effort to highlight converging results in the various experiments, be it physiological, microbial, metabolomic or transcriptomic data. This could be achieved by briefly discussing the relevance of the individual taxa/pathways/metabolites highlighted as significant in the discussion section.
- Although the use of antibiotics has been criticized, gut microbiota transfer studies after antibiotics has yielded strong, convergent phenotypes between donors and recipients. Given the observed differences in terms of aerobic/anaerobic bacteria between the obese microbiota and normal chow microbiota, would it be possible that the absence of antibiotics does not fully allow the HFD-donor microbiota to establish completely in the recipients? Would it be possible to compare the donor and recipient microbiota communities before (if samples are available) and more importantly after transfer? Is there a convergence of the recipient microbiota towards the donor microbiota?

Minor comments

- The methods section related to metabolomics, transcriptomics and statistical analysis are unclear about the methodology used for analysing multivariate data and do not refer to any multiple testing correction strategy. How were these data analysed and corrected?
- The reporting of the omics data (16s, transcriptomics, metabolomics) is under-developed. In the current state of the paper, these data do not seem to bring much evidence for the discussion.
- The role of IL17 in recipients of lean microbiota should probably be discussed in relation with the previous paper from the group (Garidou et al, Cell Metab 2015).

Reviewer #2:

Summary:

Nicolas et al. show that the transfer of dysbiotic cecal microbiota, from high fat-diet (HFD) fed mice into conventional mice, alters the gut microbiota and microbiome and reduces hepatic gluconeogenesis. This was evidenced by reduced glucose excursion in the pyruvate tolerance test (PTT) and reduced hepatic PEPCK activity. When a similar cohort of mice was fed a high-fat, very low carbohydrate diet following cecal transfer, there were also alterations in hepatic glucose metabolism, coincident with alterations in the microbiota and microbiome both in response to the diet and microbiota transfer. Results obtained using a complimentary genetically obese mouse model were largely congruent with the overall concepts.

General remarks:

The key conclusions regarding the observed phenotype (altered gluconeogenesis following gut microbiome perturbation) are convincing. While these findings are intriguing, the precise nature of the microbiome shifts in the recipient are rather descriptive, and the mechanisms by which this perturbation imparts the phenotype are not clearly identified/discussed. In addition, many areas warrant further discussion, and the conclusions drawn do not always flow from the results, though many of these can be addressed by the points below.

The paper represents a significant body of work focused on how intestinal microbiota influence host metabolism, a field that is important and rapidly evolving. The primary advance of this study is of a conceptual/technical nature: The transfer of microbiota from obese models to normal mice with an intact microbiome. This is somewhat novel and may be of relevance for translational and clinical applications. In addition, it challenges the assumption that obesity-induced gut microbial dysbiosis conveys negative metabolic health impacts.

Major Criticism:

1. A recognized challenge in studies of the microbiome is linking observed changes in microbiota/-biome to phenotypes, particularly mechanistically. Through a number of omics platforms, the authors attempt to address this, however the precise nature of the microbiome shifts, and the mechanisms by which these perturbations imparts the phenotype are not clearly identified/discussed.

- Please speculate on mechanisms by which the observed changes in the microbiome may be altering gluconeogenesis.
- In Fig.6C the differences between OM-HFD and PBS are reduced compared to those in 6B, despite the persistence of improved pyruvate tolerance. Given this, it would be informative to compare TransNC microbiome to Trans 72% HFD microbiome (similar to Fig. S8A-C), to confirm or clarify the pathways that may contribute to the gluconeogenic alterations, post HFD, as it does not appear that glyoxylate/dicarboxylate differences persist at Trans 72%HFD.
- 2. Please clarify further the nature of the alterations in hepatic glucose metabolism, to better support your title and conclusions.
 - The authors claim (line 371) that "the role of reduced fasting glycemia" in their observed phenotype has truly been "excluded." To confirm this, please include the area under the curve (AUC) for each group in all the PTT figures. Please also include detail in methods for how AUCs were calculated, i.e. what baseline was utilized (suggest utilizing timepoint-0).
 - In protocol 2, 6 weeks of 72% HFD produced a disproportionate reduction in PEPCK/G6P protein, relative to the decrease in activity (Fig. 5G-I), whereas the opposite relationship was noted in Protocol 1 when mice were fed NC (Fig. 1F-H). While there are undoubtedly alterations in pyruvate tolerance and biochemical markers of hepatic gluconeogenesis in mice transferred dysbiotic microbiota on both diets, please discuss these discrepancies and inconsistencies and perhaps elaborate on possible mechanisms responsible.
 - In addition to the effect on hepatic gluconeogenesis, it would be informative to also investigate other aspects of hepatic glucose metabolism, as well as glucose and insulin tolerance, in these dysbiosis transfer models. This is important if claiming that the "Transfer of Dysbiotic Gut Microbiota ... Prevents HFD-impaired Glucose Metabolism."
 - If possible, assessing hepatic glucose production through a hyperinsulinemic euglycemic clamp, is the gold standard measure for demonstrate these alterations.
- 3. The authors present data regarding flux through de novo lipogenesis and suggest a possible connection to gluconeogenesis (Fig 1I), but this is not adequately discussed and clarified in the results/discussion. Please address this in more detail, including possible mechanism and/or reference that link these together and allow the reader to place this data in proper context.
- 4. Regarding Fig.2, metabolomics data: please provide further clarification of methods used, particularly sample size (was there pooling?) and statistical methods. Calculation of statistical significance for lactate and pyruvate is important for the conclusions drawn using this data.
- 5. There were differences in the baseline microbiota in recipient groups prior to cecal transfer, however the authors comment that this does not affect basal glucose metabolism. Please provide some evidence of this as supplementary data, rather than 'data not shown.'
- 6. A number of measures were made in Protocol 1, which allowed a thorough interpretation of the phenotype, however many of these were not included in Protocol 2. Protocol 1 and 2 would be more comparable and better interpreted if these measures, for example, basal and fed BG, and gut and liver histology, were include in both.

Minor Criticisms:

1. In accordance with ARRIVE guidelines (Animal Research: Reporting of In Vivo Experiments), please include additional methods to describe mouse housing conditions.
2. Line 272, Nod2KO; the authors suggests their gluconeogenic phenotype is dependent on Nod2 expression. This is based on a PTT in ND fed NOD2KO mice with microbial transfers, showing no difference between groups. However, this is confusing, since previously Denou et al. 2015, showed that pyruvate tolerance was worsened in NOD2KO mice, whereas here it appears improved. Please clarify this finding and discuss further how NOD2 may provide mechanistic insight in the current model.
3. Since 'NC long term' did not show altered gluconeogenesis, please clarify text lines 293-295 to accurately reflect data shown.
4. Please show data regarding 'NC long term,' and avoid 'data not shown' wherever possible.
5. Please comment on your choice of 6 h fasting prior to PTT, as 16 h is the typical standard in the literature, as it depletes hepatic glycogen.
6. Please discuss Fig.S6E, where FFA are elevated. Particularly in relation to the findings that OM recipient mice on NC show decreased de novo lipogenic gene expression (see Fig.1J).
7. In Fig.S7 (and line 285), the time point at which these samples were obtained is unclear. Please clarify.
8. Please clarify whether Fig.S1F is in relation to the cecal contents of donors, or assessment of

feces of recipient. It appears to be the former.

9. While glycogen and PKA targets were not statistically different in Fig.1, including serum glucagon levels would more convincingly show that glucagon was not playing a role in altered hepatic gluconeogenesis.
10. Please correct Fig reference in line 273 - should be 5J.
11. Clarify y-axis label for Fig.S6B&C.
12. Please include some metabolic characteristics of lean and obese donors, for example body weight/adiposity.
13. Please include the reference gene used for qPCR in methods.
14. Line 51 - please improve word flow.
15. Line 80 - grammar.
16. Line 128 - please clarify, does not match Fig.S1C-F.
17. Line 94-95 - unclear meaning, grammar.
18. Line 129 - Fig. reference incorrect (Fig.S1E-F).
19. Line 162-163 -A more accurate and inclusive summary of the data in Fig.1 would improve this section.
20. Fig.S6, is not described in alphabetical order.
21. Line 374-5: please include a reference to substantiate your claim that the increase in plasma levels of gluconeogenic substrates suggests 'smaller utilization of the liver'.
22. Methods - please provide the macronutrient breakdown of the 60% HFD as has been done for the 72% HFD.
23. Fig.3D&G, font is different.
24. Please list the groups in the figures consistently. For example, the order of groups in Fig.1 is different to Fig4.
25. Line 283-284 - please further discuss this possible divergent phenotype.

1st Revision - authors' response

28 November 2016

Point-by-point responses to reviewers

All our corrections to the main text are underlined to help both the editor and the reviewers finding them the easiest way.

Reviewer #1:

In this manuscript, Serino and co-workers investigate an intriguing interaction between gut microbiota transfer and diet in glucose homeostasis, through a combination of physiological phenotyping, 16s rDNA pyrosequencing, metabolomics and transcriptomics. Using an antibiotics-free microbiota transfer approach, the authors obtain surprising results compared to previously reported studies. The authors observe a lower bacterial DNA load in the HFD inoculum, mostly related to a decrease of anaerobic bacteria. They also observe an increase in IL17 expression in the recipients of lean microbiota and a lower hepatic gluconeogenesis in the recipient of an obese microbiota fed a HFD, compared to recipients of lean microbiota or vehicle.

I would be grateful if the authors could take the following comments into account:

- The manuscript is generally well written and reports a wealth of data. The discussion should be tighter and focus only on the main findings to enhance clarity.

→ *We thank the reviewer for this comment. We tried to limit the discussion (page 15) matching both this comment and the comments of reviewer #2.*

- Along the same lines the authors should make an effort to highlight converging results in the various experiments, be it physiological, microbial, metabolomic or transcriptomic data. This could be achieved by briefly discussing the relevance of the individual taxa/pathways/metabolites highlighted as significant in the discussion section.

→ *We really thank the reviewer for this comment, which we believe helpful to ameliorate the discussion section. The main text (lines 441-444, page 19) has been changed as it follows:*

- Key converging results obtained by combining the two protocols presented in this study are related to the effects of HFD-microbiota on the reduction of i) fasting glycaemia and ii) markers of hepatic gluconeogenesis, and the increase of iii) the relative abundance of Firmicutes and iiiii) the glyoxylate and dicarboxylate microbial pathway on NC.

- Although the use of antibiotics has been criticized, gut microbiota transfer studies after antibiotics has yielded strong, convergent phenotypes between donors and recipients.

→ *Respectfully, we only agree in part with this comment, since the work from Ellekilde et al. clearly shows that treating the recipient with antibiotics can only provide a blunted metabolic phenotype, which is similar but not close to the one of the donors (Ellekilde M, Selfjord E, Larsen CS, Jakešević M, Rune I, Tranberg B, Vogensen FK, Nielsen DS, Bahl MI, Licht TR, Hansen AK, Hansen CH (2014) Transfer of gut microbiota from lean and obese mice to antibiotic-treated mice. Sci Rep 4: 5922). On the other hand, other studies provided different and stronger data, as this reviewer correctly said. In our study, the major point we wanted to address is to avoid any side-effect induced by antibiotics, not only the ones logically occurring on the gut microbiota of treated mice, but also effects on liver, for instance, as shown by Membrez et al. (Membrez M, Blancher F, Jaquet M, Bibiloni R, Cani PD, Burcelin RG, Corthesy I, Mace K, Chou CJ (2008) Gut microbiota modulation with norfloxacin and ampicillin enhances glucose tolerance in mice. Faseb J 22: 2416-2426).*

Given the observed differences in terms of aerobic/anaerobic bacteria between the obese microbiota and normal chow microbiota, would it be possible that the absence of antibiotics does not fully allow the HFD-donor microbiota to establish completely in the recipients?

→ *We understand this point. However, also the presence of antibiotics may not allow a full establishment of the exogenous microbiota as shown by Manichanh et al. (Manichanh C, Reeder J, Gibert P, Varela E, Llopis M, Antolin M, Guigo R, Knight R, Guarner F (2010) Reshaping the gut microbiome with bacterial transplantation and antibiotic intake. Genome Res 20: 1411-1419). This is a key study we took into account when we developed our model of transfer of gut microbiota.*

Would it be possible to compare the donor and recipient microbiota communities before (if samples are available) and more importantly after transfer? Is there a convergence of the recipient microbiota towards the donor microbiota?

→ *We have no more sample to repeat the analysis. However, this reviewer can find an attempt of answer to this point in the **Appendix Figure S1J,K**, by comparing two transplants from lean donors (like the recipient mice, which are all lean and on NC before the transfer) and HFD-induced obese mice. The differences of the two transplants are clear. Thus, based on these data and those on **Figure 4B** and **Figure 6B** we would say that there is a sort of convergence of the recipient microbiota towards the donor microbiota because mice inoculated with the HFD-microbiota show the biggest difference when compared to the other groups of inoculated mice.*

Minor comments

- The methods section related to metabolomics, transcriptomics and statistical analysis are unclear about the methodology used for analysing multivariate data and do not refer to any multiple testing correction strategy. How were these data analysed and corrected?

→ *We thank the reviewer for these points which have been ameliorated also taking into account the comments of reviewer#2. We used a pool of six mice per group. For the metabolomics analysis the figure legend and the **Materials and Methods** section have been modified as it follows (lines 819-824, page 32; and lines 510-521, page 22).*

- **Figure 2. Transfer of dysbiotic vs. eubiotic gut microbiota in NC-fed conventional mice affect serum metabolome.**

A) heat-map analysis of serum metabolome; detailed histograms for B) serum lactate and C) serum pyruvate in antibiotic-free NC-fed conventional mice inoculated with either the

vehicle (PBS) or cecal microbiota from either lean mice or HFD-fed mice (Conv + PBS, Conv + LM, Conv + OM(HFD), respectively). A pool of serum samples was used per group (n=6).

- **Metabolomics analysis.** Plasma samples (100 μ L, out of a pool of n=6 mice per group) were diluted with 600 μ L of deuterium oxide (D₂O) and centrifuged at 5,000g for 10 min before they were placed in 5 mm NMR tubes. ¹H NMR spectra were obtained on a Bruker DRX-600 Avance NMR spectrometer operating at 600,13 MHz for ¹H resonance frequency using an inverse detection 5mm 1H-13C-15N cryoprobe attached to a CryoPlatform (the preamplifier cooling unit). The ¹H NMR spectra were acquired at 300K using the Carr-Purcell-Meiboom Gill (CPMG) spin-echo pulse sequence with pre-saturation, with a total spin-echo delay (2nt) of 64 ms to attenuate broad signals from proteins and lipoproteins. A total of 128 transients were collected into 32k data points using a spectral width of 12 ppm, a relaxation delay of 5 s and an acquisition time of 2.28s. Prior to Fourier Transformation, an exponential line broadening function of 0.3 Hz was applied to the FID. NMR spectra were phased and baseline corrected, then metabolites signals were integrated, and normalized to the total spectral area.

As for the microarray, we have added the following sentence (lines 538-541, page 23):

- Genes were considered differently expressed between Conv + OM(HFD) vs. Conv + PBS and between Conv + LM vs. Conv + PBS groups when $p < 0.05$. We also considered a logarithm of the fold change vs. Conv + PBS between -2.2 (for down-stream regulation) and 2.5 (for up-stream regulation).

- The reporting of the omics data (16s, transcriptomics, metabolomics) is under-developed. In the current state of the paper, these data do not seem to bring much evidence for the discussion.

→ *The reviewer is right, we have privileged the metabolic phenotype based on hepatic gluconeogenesis modulation. However, by answering to the second point raised by this reviewer we hope we ameliorated the discussion and underlined the role of Firmicutes in the two protocols of the study.*

- The role of IL17 in recipients of lean microbiota should probably be discussed in relation with the previous paper from the group (Garidou et al, Cell Metab 2015).

→ *We thank the reviewer for this point, which ameliorated our discussion. We have modified the main text as it follows (lines 400 to 405, page 17):*

- This may result in the elaboration of an efficient immune response, as suggested by the high significant induction of IL-17a in the intestine of mice inoculated with lean microbiota. By contrast, HFD-microbiota did not induce this raise, in accordance with our previous report (Garidou et al, 2015). Therefore, the efficient immune response may have a systemic beneficial impacts such as the ones observed on the liver and the white adipose tissue.

We really hope Reviewer #1 will appreciate our answers to his/her comments and the modifications we made accordingly to improve our manuscript.

Reviewer #2:

Summary:

Nicolas et al. show that the transfer of dysbiotic cecal microbiota, from high fat-diet (HFD) fed mice into conventional mice, alters the gut microbiota and microbiome and reduces hepatic gluconeogenesis. This was evidenced by reduced glucose excursion in the pyruvate tolerance test (PTT) and reduced hepatic PEPCK activity. When a similar cohort of mice was fed a high-fat, very low carbohydrate diet following cecal transfer, there were also alterations in hepatic glucose metabolism, coincident with alterations in the microbiota and microbiome both in response to the diet and microbiota transfer. Results obtained using a complimentary genetically obese mouse model were largely congruent with the overall concepts.

General remarks:

The key conclusions regarding the observed phenotype (altered gluconeogenesis following gut microbiome perturbation) are convincing. While these findings are intriguing, the precise nature of the microbiome shifts in the recipient are rather descriptive, and the mechanisms by which this perturbation imparts the phenotype are not clearly identified/discussed. In addition, many areas warrant further discussion, and the conclusions drawn do not always flow from the results, though many of these can be addressed by the points below.

The paper represents a significant body of work focused on how intestinal microbiota influence host metabolism, a field that is important and rapidly evolving. The primary advance of this study is of a conceptual/technical nature: The transfer of microbiota from obese models to normal mice with an intact microbiome. This is somewhat novel and may be of relevance for translational and clinical applications. In addition, it challenges the assumption that obesity-induced gut microbial dysbiosis conveys negative metabolic health impacts.

Major Criticism:

1. A recognized challenge in studies of the microbiome is linking observed changes in microbiota/-biome to phenotypes, particularly mechanistically. Through a number of omics platforms, the authors attempt to address this, however the precise nature of the microbiome shifts, and the mechanisms by which these perturbations impart the phenotype are not clearly identified/discussed. - Please speculate on mechanisms by which the observed changes in the microbiome may be altering gluconeogenesis.

→ *We thank the reviewer for this point, which we believe it improves the discussion section. The main text has been improved as it follows (lines 383 to 391, pages 16 to 17):*

- Indeed, in both protocols used, the transfer of gut microbiota induced a significant increase of Firmicutes, which are Gram positive bacteria harbouring a more developed peptidoglycan when compared to Gram negative bacteria. Of note, peptidoglycan represents a ligand for NOD2, which may provide a mechanism of action for the modulation of hepatic gluconeogenesis. However, data in this study do not match the NOD2 KO-induced worsening of pyruvate-tolerance observed in the work cited above (Denou et al, 2015). Nonetheless, the diverse gut microbiota harboured by NOD2 KO mice, as shown by us (Denou et al, 2015) and others (Mondot et al, 2012), may contribute to blunt the aforementioned change in hepatic glucose production induced by the inoculation.

- In Fig.6C the differences between OM-HFD and PBS are reduced compared to those in 6B, despite the persistence of improved pyruvate tolerance. Given this, it would be informative to compare TransNC microbiome to Trans 72% HFD microbiome (similar to Fig. S8A-C), to confirm or clarify the pathways that may contribute to the gluconeogenic alterations, post HFD, as it does not appear that glyoxylate/dicarboxylate differences persist at Trans 72%HFD.

→ *We fully agree with the reviewer about this very good point. In fact, as shown in **Figure 6K,L**, glyoxylate and dicarboxylate microbial pathway is no longer affected when mice are fed a 72%HFD. Therefore, it may count for the NC-feeding period (Trans_NC) but not for the 72%HFD (Trans_72%HFD). Based on this important point raised by the reviewer, we have modified the main text as it follows (lines 425 to 432, page 18):*

- *Moreover, the significant negative correlation between the glyoxylate and dicarboxylate microbial pathway and the index of pyruvate-tolerance (AUC) suggests a link between this microbial activity and the regulation of hepatic glucose production. Our hypothesis is supported by a recent publication which observed that the glyoxylate and dicarboxylate microbial pathway is among the most affected in the model of Zucker diabetic fatty rats (Dong et al, 2016). Therefore, targeting microbial genes involved in this pathway may be effective for the control of hepatic function, but only during the transfer on NC. In fact, this pathway is no longer affected after the switch on 72%HFD, suggesting a different microbial regulation of hepatic glucose production.*

→ We also provide below the cladogram comparisons per group, as requested by the reviewer, just to show that changes are consistent with those reported in **Figure 6B**. In fact, for mice treated with PBS and mice inoculated with ob-microbiota, gut microbiota changes for Trans_NC time-point are closer each other (first two cladograms) than with the gut microbiota from mice inoculated with HFD-microbiota (third cladogram). This demonstrates that the inoculation of HFD-microbiota had a major impact on the gut microbiota of recipient mice than ob-microbiota.

Cladogram 1

Cladogram 2

Cladogram 3

2. Please clarify further the nature of the alterations in hepatic glucose metabolism, to better support your title and conclusions.

- The authors claim (line 371) that "the role of reduced fasting glycemia" in their observed phenotype has truly been "excluded." To confirm this, please include the area under the curve (AUC) for each group in all the PTT figures. Please also include detail in methods for how AUCs were calculated, i.e. what baseline was utilized (suggest utilizing timepoint-0).

→ *We thank the reviewer for this point. All AUCs have been provided for PTTs (and also IPGTT and OGTT) and added as inset. We calculated the AUC as the sum of glycaemic values (in mmol/L) from 0 to 120 min, as suggested by this reviewer, and divided it by 120 (total minutes from 0 to 120 time-point) and multiplied by 1000 to show it as $\mu\text{mol/L} \times \text{min}$. We have added this description in the Appendix Supplementary Methods, chapter "Intraperitoneal Pyruvate-(IPPTT), Glucose-(OGTT) and Glucose or Insulin-(IPITT) oral (O) or intraperitoneal (IP) tolerance tests." as it follows:*

- Area under the curve (AUC) is also shown as inset for IPPTTs and IPGTTs/OGTT. AUC has been calculated as the sum of glycaemic values (in mmol/L) out of the IPPTT from 0 to 120 min (from -30 to 120 min for IPGTT/OGTT), divided by 120 (total minutes from 0 to 120 time-point) and multiplied by 1000 to show it as $\mu\text{mol/L} \times \text{min}$.

This calculation has been validated in Serino et al., Gut, 2012.

- In protocol 2, 6 weeks of 72% HFD produced a disproportionate reduction in PEPCK/G6P protein, relative to the decrease in activity (Fig. 5G-I), whereas the opposite relationship was noted in Protocol 1 when mice were fed NC (Fig. 1F-H). While there are undoubtedly alterations in pyruvate tolerance and biochemical markers of hepatic gluconeogenesis in mice transferred dysbiotic microbiota on both diets, please discuss these discrepancies and inconsistencies and perhaps elaborate on possible mechanisms responsible.

→ *We fully agree with this point. We have modified the main text as it follows (lines 369 to 370, page 16):*

- Therefore, the nutritional status may account for discrepancies observed in protocol#2 between the amount and the activity of PEPCK, as suggested by evidence in the literature (Chen et al, 2016).

- In addition to the effect on hepatic gluconeogenesis, it would be informative to also investigate other aspects of hepatic glucose metabolism, as well as glucose and insulin tolerance, in these dysbiosis transfer models. This is important if claiming that the "Transfer of Dysbiotic Gut Microbiota ... Prevents HFD-impaired Glucose Metabolism."

→ *We strongly agree with this comment and we have already investigated, where possible, effects on glucose- (as IPGTT or OGTT) and insulin-tolerance in our new model. Data are now provided within Appendix Figures. We have added the following:*

- The OGTT for protocol #1 as **Appendix Fig.S2G** plus AUC as inset
- The IPGTT on NC for protocol #2 as **Appendix Fig.S5H** plus AUC as inset
- The IPITT on NC for protocol #2 as **Appendix Fig.S5I**
- The IPGTT on 72%HFD for protocol #2 as **Appendix Fig.S5J** plus AUC as inset
- The IPITT on 72%HFD for protocol #2 as **Appendix Fig.S5K**

- If possible, assessing hepatic glucose production through a hyperinsulinemic euglycemic clamp, is the gold standard measure for demonstrate these alterations.

→ *We strongly agree with this comment. However, we could not proceed to this investigation, in this model, because of the heavy surgical intervention that it requires; thus, this intervention may induce a local inflammation in the site of the intra-catheterization and hence, change the systemic inflammatory tone, which could bias our model towards the transfer of gut microbiota.*

3. The authors present data regarding flux through de novo lipogenesis and suggest a possible connection to gluconeogenesis (Fig 1I), but this is not adequately discussed and clarified in the results/discussion. Please address this in more detail, including possible mechanism and/or reference that link these together and allow the reader to place this data in proper context.

→ We thank the reviewer for this point. We have changed the main text according to this comment, as it follows (lines 408 to 410, pages 17 to 18):

- Evidences in the literature suggest that de novo lipogenesis may be a qualitative marker for carbohydrate intake in humans, which is directly related to hepatic glucose production (Schwarz et al, 1995)

4. Regarding Fig.2, metabolomics data: please provide further clarification of methods used, particularly sample size (was there pooling?) and statistical methods. Calculation of statistical significance for lactate and pyruvate is important for the conclusions drawn using this data.

→ We thank the reviewer for this point, allowing clarifying our manuscript. We used a pool of six mice per group. The figure legend and the **Materials and Methods** section have been modified as it follows (lines 819-824, page 32; and lines 510-521, page 22):

- Figure 2. Transfer of dysbiotic vs. eubiotic gut microbiota in NC-fed conventional mice affect serum metabolome.

A) heat-map analysis of serum metabolome; detailed histograms for B) serum lactate and C) serum pyruvate in antibiotic-free NC-fed conventional mice inoculated with either the vehicle (PBS) or cecal microbiota from either lean mice or HFD-fed mice (Conv + PBS, Conv + LM, Conv + OM(HFD), respectively). A pool of serum samples was used per group (n=6).

- **Metabolomics analysis.** Plasma samples (100 μ L, out of a pool of n=6 mice per group) were diluted with 600 μ L of deuterium oxide (D₂O) and centrifuged at 5,000g for 10 min before they were placed in 5 mm NMR tubes. ¹H NMR spectra were obtained on a Bruker DRX-600 Avance NMR spectrometer operating at 600,13 MHz for ¹H resonance frequency using an inverse detection 5mm 1H-13C-15N cryoprobe attached to a CryoPlatform (the preamplifier cooling unit). The ¹H NMR spectra were acquired at 300K using the Carr-Purcell-Meiboom-Gill (CPMG) spin-echo pulse sequence with pre-saturation, with a total spin-echo delay (2nt) of 64 ms to attenuate broad signals from proteins and lipoproteins. A total of 128 transients were collected into 32k data points using a spectral width of 12 ppm, a relaxation delay of 5 s and an acquisition time of 2.28s. Prior to Fourier Transformation, an exponential line broadening function of 0.3 Hz was applied to the FID. NMR spectra were phased and baseline corrected, then metabolites signals were integrated, and normalized to the total spectral area.

5. There were differences in the baseline microbiota in recipient groups prior to cecal transfer, however the authors comment that this does not affect basal glucose metabolism. Please provide some evidence of this as supplementary data, rather than 'data not shown.'

→ We have added the supporting data as **Appendix Fig.S1A-E**.

6. A number of measures were made in Protocol 1, which allowed a thorough interpretation of the phenotype, however many of these were not included in Protocol 2. Protocol 1 and 2 would be more comparable and better interpreted if these measures, for example, basal and fed BG, and gut and liver histology, were include in both.

→ We agree with this reviewer that having the same parameters would allow a better comparison of the two protocols. However, we intended to compare only the major phenotype, i.e. hepatic glucose production and its markers (G6Pase and PEPCK). Then, the additional parameters that are presented in each protocol are strictly related to the protocol itself. For example, FED glycaemia only appears in the second protocol because it is a key parameter reflecting the impact of hepatic gluconeogenesis, given the very low carbohydrate diet we used (72%HFD) in protocol#2.

Therefore, for protocol#2, we just confirmed the observed reduction in pyruvate-tolerance on NC, as shown in **Figure 5B**, before to challenge mice with the 72%HFD.

Minor Criticisms:

1. In accordance with ARRIVE guidelines (Animal Research: Reporting of In Vivo Experiments), please include additional methods to describe mouse housing conditions.

→ We thank the reviewer for this point. We downloaded the ARRIVE guidelines pdf (<https://www.nc3rs.org.uk/sites/default/files/documents/Guidelines/NC3Rs%20ARRIVE%20Guidelines%202013.pdf>) and changed the **Materials and Methods** section chapter "**Animal model and diet**" accordingly, as it follows (line 457, page 20):

- **Animal model and diet.** 6-wk-old C57Bl/6 male mice (Charles River, L'Arbresle, France) were fed a normal chow (NC) for 4 weeks (protocol #1) or a NC and then a high-fat diet (HFD) (~72% fat (corn-oil and lard), 28% protein and <1% carbohydrate, SAFE, Augy, France)(Serino et al, 2012b) for six weeks (protocol #2) ENREF 18. Mice were group-housed (five-six mice per cage) in a specific-pathogen free controlled environment (inverted 12-h daylight-cycle, light off at 10:00 a.m.). Six-hour-fasted mice were sacrificed by cervical dislocation. Then, tissues were collected and snap-frozen in liquid nitrogen. All animal experimental procedures were approved by the local ethical committee of Rangueil University Hospital (Toulouse, France).

2. Line 272, Nod2KO; the authors suggest their gluconeogenic phenotype is dependent on Nod2 expression. This is based on a PTT in ND fed NOD2KO mice with microbial transfers, showing no difference between groups. However, this is confusing, since previously Denou et al. 2015, showed that pyruvate tolerance was worsened in NOD2KO mice, whereas here it appears improved. Please clarify this finding and discuss further how NOD2 may provide mechanistic insight in the current model.

→ We agree with this comment. The discussion has been modified as it follows (line 379 to 391, pages 16 to 17):

- Interestingly, the aforementioned hepatic phenotypes were blunted in NOD2 KO mice, previously reported by our group to develop features of metabolic syndrome on NC and to a greater extent on a HFD (Denou et al, 2015). This result suggests the involvement of the NOD2 microbial sensor in the management of the metabolic effects induced by the transfer of gut microbiota in recipient mice. Indeed, in both protocols used, the transfer of gut microbiota induced a significant increase of Firmicutes, which are Gram positive bacteria harbouring a more developed peptidoglycan when compared to Gram negative bacteria. Of note, peptidoglycan represents a ligand for NOD2, which may provide a mechanism of action for the modulation of hepatic gluconeogenesis. However, data in this study do not match the NOD2 KO-induced worsening of pyruvate-tolerance observed in the work cited above (Denou et al, 2015). Nonetheless, the diverse gut microbiota harboured by NOD2 KO mice, as shown by us (Denou et al, 2015) and others (Mondot et al, 2012), may contribute to blunt the aforementioned change in hepatic glucose production induced by the inoculation.

3. Since 'NC long term' did not show altered gluconeogenesis, please clarify text lines 293-295 to accurately reflect data shown.

→ We thank the reviewer for ameliorating this point. The main text has been changed accordingly, as it follows (lines 296 to 298, page 13):

- Altogether, these data show that mice inoculated with either HFD- or ob-microbiota had acutely lower hepatic gluconeogenesis whether they were fed NC or 72%HFD, lower white adipose cell size and a slight intestinal inflammation with no change in intestinal permeability.

4. Please show data regarding 'NC long term,' and avoid 'data not shown' wherever possible.

→ *We managed this point by removing the terms “not shown” and by adding the related IPPTT and its AUC as inset in **Appendix Fig.S5L** and also wherever needed.*

5. Please comment on your choice of 6 h fasting prior to PTT, as 16 h is the typical standard in the literature, as it depletes hepatic glycogen.

→ *Mice were fasted for 6 hours because of the inverted light cycle (inverted 12-h daylight-cycle, light off at 10:00 a.m.). Therefore, we fasted the mice during their light period, in which they eat the less, as confirmed by the overall low hepatic glycogen content for all groups, as shown in **Fig.1D**.*

6. Please discuss Fig.S6E, where FFA are elevated. Particularly in relation to the findings that OM recipient mice on NC show decreased de novo lipogenic gene expression (see Fig.1J).

→ *We took into account this point. The main text has been changed as it follows (line 405 to 408, page 18):*

- With regard to the latter, in mice inoculated with the HFD-microbiota the reduced expression of de novo lipogenic genes is in accordance and may be an upstream event leading to smaller adipocytes and increased serum FFA, as observed in axenic mice (Backhed et al, 2004).

7. In Fig.S7 (and line 285), the time point at which these samples were obtained is unclear. Please clarify.

→ *We took into account this point. The main text has been changed as it follows (lines 288 to 295, pages 12 to 13):*

*- We also investigated whether the different origins of dysbiotic gut microbiota may affect intestinal inflammation and permeability. As shown in **Appendix Fig.S7A**, at the end of protocol#2, inoculation with HFD-microbiota induced a significant increase in the ileum of iNOS, IFN γ and IL-6 gene expression and a tendency to increase the majority of the analysed inflammatory markers, also shown in mesenteric lymph-nodes (**Appendix Fig.S7B**), whereas the ob-microbiota transfer did not significantly affect these parameters. Neither dysbiotic gut microbiota significantly changed the expression of tight junction proteins (**Appendix Fig.S7C**), whereas both transfers induced a tendency to increase defensins production (**Appendix Fig.S7D**).*

8. Please clarify whether Fig.S1F is in relation to the cecal contents of donors, or assessment of feces of recipient. It appears to be the former.

→ ***Appendix Fig.S1** has been modified according to this point by adding a title in **Appendix Fig.S1F-K**.*

9. While glycogen and PKA targets were not statistically different in Fig.1, including serum glucagon levels would more convincingly show that glucagon was not playing a role in altered hepatic gluconeogenesis.

→ *We agree with this reviewer that adding the glucagon serum levels would strengthen the message of the figure. However, we privileged the down-stream effects of glucagon. In fact, even with significant different serum glucagon levels, we could not be sure that the glucagon signalling would be affected. Thus, we opted for a more interpretable data-set, to save serum for further analyses, such as metabolomics one.*

10. Please correct Fig reference in line 273 - should be 5J.

→ *Done, we are sorry for the mistake.*

11. Clarify y-axis label for Fig.S6B&C.

→ Done, “g” was specified as “gram”.

12. Please include some metabolic characteristics of lean and obese donors, for example body weight/adiposity.

→ We have added these data to **Appendix Fig.S1A,E**.

13. Please include the reference gene used for qPCR in methods.

→ We added the following sentence in the “Materials and Methods” section, chapter “**RNA extraction and qPCR in liver, ileum and MLN**”. The text has been modified as it follows (line 554, page 24):

- The housekeeping gene used in this study is the Ribosomal Protein L19 (RPL19).

14. Line 51 - please improve word flow.

→ We have changed the main text as it follows (lines 50 to 51, page 2):

Our findings provide a new perspective on gut microbiota dysbiosis, potentially useful to better understand the aetiology of metabolic diseases.

15. Line 80 - grammar.

→ We have changed the main text as it follows (line 79, page 3):

- These mice enabled to uncover few molecular mechanisms by which gut microbiota modulates host metabolism (Backhed et al, 2007)

16. Line 128 - please clarify, does not match Fig.S1C-F.

→ We are sorry for the mistake. The main text has been changed as it follows (lines 128 to 130, page 5):

- By contrast, the two transplants from the same donor showed a strong homogeneity after one week (**Appendix Fig.S1J-K**). (We also modified **Appendix Fig.S1J-K**, as shown).

17. Line 94-95 - unclear meaning, grammar.

→ A trait was missing; we are sorry for the mistake. The main text has been changed as it follows (lines 92 to 95, page 4):

- We found that transfer of dysbiotic gut microbiota to conventional mice acutely reduces markers of hepatic gluconeogenesis during normal chow and protects towards high-fat diet-increased markers of hepatic gluconeogenesis and adiposity, together with changes in both gut microbiota and microbiome.

18. Line 129 - Fig. reference incorrect (Fig.S1E-F).

→ We have corrected as stated in our response to comment #16.

19. Line 162-163 -A more accurate and inclusive summary of the data in Fig.1 would improve this section.

→ We thank the reviewer for this advice. The main text has been changed as it follows (lines 162 to 164, page 7):

- Altogether, these data show that the transfer of HFD-microbiota lowered fasting glycaemia and markers of hepatic gluconeogenesis in association with a reduced gluconeogenic enzyme activity, without affecting neither glucagon signalling nor hepatic glycogen content.

20. Fig.S6, is not described in alphabetical order.

→ We are sorry for this mistake. The main text has been changed as it follows (lines 277 to 280, page 12):

- Despite a lack of significant change in body weight (Appendix Fig.S6A), fat (Appendix Fig.S6B) and lean mass (Appendix Fig.S6C) mice inoculated with the HFD-microbiota displayed significant smaller adipocytes (Appendix Fig.S6D) when compared to control mice.

21. Line 374-5: please include a reference to substantiate your claim that the increase in plasma levels of gluconeogenic substrates suggests 'smaller utilization of the liver'.

→ We added the following reference (line 374, page 16):

- Gluconeogenesis and ketogenesis in perfused liver of rats submitted to short-term insulin-induced hypoglycaemia from (Albuquerque et al, 2008).

22. Methods - please provide the macronutrient breakdown of the 60% HFD as has been done for the 72% HFD.

→ We have added the requested informations as it follows (line 466, page 20):

- Protocol #1. Donor mice: 8-wk-old C57Bl/6 male mice (Charles River, L'Arbresle, France) were either fed a 60% fat HFD (60% fat, 20% carbohydrates, 20% proteins) (Serino et al, 2007) or a NC for three months

23. Fig.3D&G, font is different.

→ We have verified this point and for both Fig.3D&G font is Arial 12.

24. Please list the groups in the figures consistently. For example, the order of groups in Fig.1 is different to Fig4.

→ We agree with the reviewer that the list is not the same. The software used does not allow a different listing, whatever the list you provide it. We made the maximum effort by matching the group colours.

25. Line 283-284 - please further discuss this possible divergent phenotype.

→ We thank the reviewer for this point. We have improved the main text as it follows (line 286, page 12):

- This result suggests that dysbiosis of gut microbiota may have a divergent metabolic impact, either smaller or no change in adipocyte size, on the white adipose tissue depending on its origin (i.e. nutritional vs. genetic).

We really hope Reviewer #2 will appreciate our answers to his/her comments and the modifications we made accordingly to improve our manuscript.

Thank you sending us your revised manuscript. We have now heard back from reviewer #2 who was asked to evaluate your study. As you will see below, the reviewer raises some remaining concerns, which we would ask you to address in a revision.

As the referee mentions, the manuscript would benefit from language and text editing, and we would strongly recommend that you have the manuscript edited by a native English speaker.

We would ask you to address some editorial issues listed below.

REFEREE REPORTS

Reviewer #2:

Response to Rebuttal

The author's resubmitted manuscript addressed many of both reviewer's criticisms and includes helpful additional data and methods. Overall this represents an improvement from the first submission, and the paper remains strong with interesting findings. However, to our judgement, several of the criticisms from both reviewers were not fully addressed. Importantly, the manuscript would benefit from a tighter and more focused discussion, highlighting converging results in the various experiments, be it physiological, microbial, metabolomic or transcriptomic data, and clearly explaining potential mechanisms. In addition, this resubmission is less polished in terms of grammar and structure.

Addressing the issues listed below, including improving the language and flow, will strengthen the manuscript and ensure that the important findings are conveyed.

1. The speculated mechanisms remain unclear and associative. Please clearly connect the stimulation of NOD2 by Firmicutes-derived peptidoglycans, and alterations in glyoxylate and dicarboxylate pathways, to hepatic gluconeogenesis (specific pathways, appropriate references).
2. The added cladograms were of interest. However, please utilize PCA analysis (similar to that depicted in the S8A-C, D-I), to identify microbiome and microbiota that are common to TransNC and Trans72% HFD states (when PTT is altered/improved) but are not present in the baseline ("B") state, as this may shed light on what microbiome/-biota alterations are playing a role in altered glucose metabolism. Please include this data/figure in the manuscript.
3. The included method of calculating AUC is not standard in the literature, and does not take into account the differences in basal blood sugar. Please provide AUCs by a standard method, (see Floch et al. "Blood glucose area under the curve: methodological aspects," Diabetes Care, 1990). Please ensure that the interpretation of results takes into account these findings, i.e. discussing non-statistically significant PTT results.
4. Regarding PEPCK protein and mRNA levels: Please discuss and include references that suggest how contrasting relationships between protein quantity and activity may be observed in the same phenotype. One would presume the same mechanism of action of "dysbiotic transfer" would be responsible in both protocol 1 and 2. (Chen et al only suggest that HFD alters PEPCK expression).
5. While carbohydrate intake is indeed connected to de novo lipogenesis, it is not clear how hepatic gluconeogenesis is directly connected to de novo lipogenesis. In fact, the two typically occur in different metabolic states. Please explain and clarify this relationship with appropriate references.
6. Regardless of the circadian phase, the duration of fasting remains less than the standard for most studies using PTT. Furthermore, the hepatic glycogen values in Fig 1D are extremely low compared to publications which 16 hours are used.
 - a. Please further explain your rationale for utilizing this limited fast, and include relevant references.

b. Please specify in the methods the conditions at which the liver was collected for the glycogen measurement.

7. Regarding line 413-316: please clearly reconcile the counterintuitive reduction in de novo lipogenic gene expression with increased serum FFA. Backhed et al (2004) found that serum triglycerides and liver FFA were similarly affected by the microbiome, not in opposite directions.

8. Lines 381-383: Albuquerque et al utilize an in situ model, focused on defining "maximal liver capacity" for gluconeogenesis. If you are suggesting that your PTT data and gluconeogenic substrate (metabolomic) data are connected by such a relationship, please discuss this proposed mechanism clearly in the text, and provide at least one additional, highly-relevant reference to substantiate this.

9. Line 81-82: the grammar remains incorrect

10. The sentence in lines 301-303 is still not entirely clear. Consider "...ob-microbiota had lower hepatic gluconeogenesis both following acute NC diet and 72%HFD."

2nd Revision - authors' response

25 January 2017

All our corrections to the main text are underlined to help both the editor and the reviewers finding them the easiest way.

Editor:

As the referee mentions, the manuscript would benefit from language and text editing, and we would strongly recommend that you have the manuscript edited by a native English speaker.

→ *We have sent the manuscript for English editing, as suggested.*

- We would also encourage you to include the source data for figure panels showing essential quantitative information or immunoblots. More information on source data can be found in our Author Guidelines: < <http://msb.embopress.org/authorguide#expandedview>.

→ *We provide source data for immunoblots of both Figure 1 and Figure 6 (please note that original titles and dates of acquisition are also provided for each part of figure) as well as the metabolomics spectra and raw data for Figure 2.*

Reviewer #2:

Response to Rebuttal

The author's resubmitted manuscript addressed many of both reviewer's criticisms and includes helpful additional data and methods. Overall this represents an improvement from the first submission, and the paper remains strong with interesting findings. However, to our judgement, several of the criticisms from both reviewers were not fully addressed. Importantly, the manuscript would benefit from a tighter and more focused discussion, highlighting converging results in the various experiments, be it physiological, microbial, metabolomic or transcriptomic data, and clearly explaining potential mechanisms.

→ *We thank the reviewer for his/her suggestions. Now, we provide a tighter (from 4 to 3 pages) and more focused discussion.*

In addition, this resubmission is less polished in terms of grammar and structure.

→ *We are sorry for that. As mentioned above, the manuscript has been sent for English editing. We managed avoiding repetition of citation for figures and we removed not useful additional wording, mostly at the beginning of each sentence. As suggested, we had to improve the flow. We believe and hope that now we have met all of these needs.*

Addressing the issues listed below, including improving the language and flow, will strengthen the manuscript and ensure that the important findings are conveyed.

→ *We greatly appreciate the general positive attitude and all of the suggestions proposed by the reviewer and Editor to ameliorate our manuscript. We really hope that we have met all of your points now.*

1. The speculated mechanisms remain unclear and associative.

→ *We really did our best to improve discussion and look for high-relevant references to reinforce our conclusions; however, we also find high difficult to find references which are appropriate for speculating mechanisms. We really hope the reviewer will appreciate our effort.*

Please clearly connect the stimulation of NOD2 by Firmicutes-derived peptidoglycans, and alterations in glyoxylate and dicarboxylate pathways, to hepatic gluconeogenesis (specific pathways, appropriate references).

→ *We understand this point but we honestly admit that a reference clearly linking all of these pathways was not found. We believe this is due in part to the novelty of our findings and in part to the fact that speculations, by definition, are hard to sustain. However, to try to meet this request the best way, we did the effort to speculate on the systemic regulation we show. We believe the references we have used (page 16) are the best we could find to provide a mechanistic explanation. We really hope this reviewer will appreciate our effort.*

2. The added cladograms were of interest. However, please utilize PCA analysis (similar to that depicted in the S8A-C, D-I), to identify microbiome and microbiota that are common to TransNC and Trans72% HFD states (when PTT is altered/improved) but are not present in the baseline ("B") state, as this may shed light on what microbiome/-biota alterations are playing a role in altered glucose metabolism. Please include this data/figure in the manuscript.

→ *We thank the reviewer for this suggestion. Accordingly, we have created the **Appendix Figure S9**, comparing TransNC and Trans72% HFD states, the way the reviewer asked. We also discuss this new figure in the main-text.*

3. The included method of calculating AUC is not standard in the literature, and does not take into account the differences in basal blood sugar. Please provide AUCs by a standard method, (see Floch et al. "Blood glucose area under the curve: methodological aspects," Diabetes Care, 1990). Please ensure that the interpretation of results takes into account these findings, i.e. discussing non-statistically significant PTT results.

→ *We thank the reviewer for providing this article, which has been added to the section Materials and Methods (page 21). Based on it, we calculated AUCs by graphpad prism, which uses the trapezoidal rule as can be seen at this link <http://www.graphpad.com/support/faqid/82/>). Of note, the trapezoidal rule method is validated by the article provided by the reviewer as stated "...variations related to the method used in estimating AUC are not clinically relevant and that a simple method such as trapezoidal rule can be used." As this reviewer suggested and as we already did accordingly, we calculated AUCs from the "0" time-point. Moreover, we discuss these new results according to the new AUCs calculations (page 6) as it follows:*

- *Moreover, since the area under the curve shows a not significant HFD-microbiota effect (Fig.1E), also the fasting glycaemia accounts for the observed trend of reduced hepatic gluconeogenesis.*

4. Regarding PEPCK protein and mRNA levels: Please discuss and include references that suggest how contrasting relationships between protein quantity and activity may be observed in the same phenotype. One would presume the same mechanism of action of "dysbiotic transfer" would be responsible in both protocol 1 and 2. (Chen et al only suggest that HFD alters PEPCK expression).

→ We agree with this comment about the fact that the work from Chen et al. does not totally help explaining the discrepancies observed for PEPCK regulation during the dysbiotic transfer. Therefore, we removed this reference.

By contrast, we believe that the nutritional status (HFD feeding) may account for these discrepancies, majorly by impacting the gut microbiota and the consequent related systemic effects. Thus, even if the phenotype is similar as obtained by transferring a dysbiotic gut microbiota, the nutritional status may be crucial and may differently affect the mechanism of action. Hence, the text (page 16) has been modified as it follows:

- The 72%HFD may account for discrepancies observed for PEPCK regulation during the dysbiotic transfer, most likely by affecting the gut microbiota of the recipient, leading to the consequent systemic effects.

5. While carbohydrate intake is indeed connected to de novo lipogenesis, it is not clear how hepatic gluconeogenesis is directly connected to de novo lipogenesis. In fact, the two typically occur in different metabolic states. Please explain and clarify this relationship with appropriate references.

→ We totally agree with this comment. We did not mean that hepatic gluconeogenesis is directly connected to de novo lipogenesis. As clearly stated by this reviewer, the two pathways typically occur in different metabolic states.

We reported in the text (page 6) that “Since among the 1021 genes significantly modulated by HFD-microbiota none of them was directly implicated in gluconeogenesis, this suggests that the decrease of markers of hepatic glucose production observed above is not due to a change in gene expression”. By this sentence, we just meant that only the inoculation with HFD-microbiota could affect genes involved in metabolic pathways, but that these genes do not relate to the hepatic glucose production. Therefore, we removed this part to avoid misinterpretation.

6. Regardless of the circadian phase, the duration of fasting remains less than the standard for most studies using PTT. Furthermore, the hepatic glycogen values in Fig 1D are extremely low compared to publications which 16 hours are used.

a. Please further explain your rationale for utilizing this limited fast, and include relevant references.

→ Given the fact that our mice are on an inverted light cycle (inverted 12-h daylight-cycle, light off at 10:00 a.m.), we fasted the mice during their light period, in which they eat the less. Therefore, this may explain the low glycogen values observed in our study. Moreover, beyond the value per se, we judged important to measure hepatic glycogen to exclude significant differences between the groups, which to us is more important than the precise value of glycogen content.

This limited fast is also used in the article from Ribeiro TA et al. (Toxicology, 2016), which has been added to the main-text (page 21) as it follows:

“Since mice were on an inverted light-cycle, IPPTT was performed by injecting pyruvate (2 g/kg) in six-hour-fasted mice (Ribeiro et al, 2016).”

b. Please specify in the methods the conditions at which the liver was collected for the glycogen measurement.

→ We have added, to the “**Hepatic glycogen dosage**” section (page 20), the following: “...from six-hour-fasted mice...”.

7. Regarding line 413-316: please clearly reconcile the counterintuitive reduction in de novo lipogenic gene expression with increased serum FFA. Backhed et al (2004) found that serum triglycerides and liver FFA were similarly affected by the microbiome, not in opposite directions.

→ We thank the reviewer for this comment. We have integrated the work from Eissing et al. (nat comm, 2013) in which the authors showed that lipogenic enzymes are upregulated in the liver of obese patients (therefore in association with bigger adipocytes, due to obesity). Moreover, Backhed et al (2004) found increased FFA in the serum of axenic mice, which is in association with the smaller adipocytes of this animal model. Therefore, we modified the text (page 17) as it follows:

- With regard to the latter, in mice inoculated with the HFD-microbiota the reduced expression of de novo lipogenic genes is in accordance with smaller adipocytes. In

fact, Eissing et al. showed that lipogenic enzymes are upregulated in the liver of obese patients (Eissing et al, 2013), and in our study lipogenic enzymes are downregulated in the liver of mice showing smaller adipocytes. Increased serum FFA were associated to smaller adipocytes in axenic mice (Backhed et al, 2004) too, in accordance with our data.

8. Lines 381-383: Albuquerque et al utilize an in situ model, focused on defining "maximal liver capacity" for gluconeogenesis. If you are suggesting that your PTT data and gluconeogenic substrate (metabolomic) data are connected by such a relationship, please discuss this proposed mechanism clearly in the text, and provide at least one additional, highly-relevant reference to substantiate this.

→ *By using this reference, we only meant to show that a biological relationship is established between hepatic glucose production regardless of the model employed. To avoid misinterpretation and avoid adding complex mechanisms, this reference was replaced (page 16) with the article in Nature by Madiraju et al. (2014) in which authors show that increased lactate plasma levels are in accordance with the metformin-reduced hepatic glucose production. Overall, the text has been rephrased according to an extensive revision of the discussion section, as requested.*

9. Line 81-82: the grammar remains incorrect

→ *We are not sure we really found the grammar error, but English language and text have been totally revised, as asked. Thus, we believe and hope that the error is corrected now.*

10. The sentence in lines 301-303 is still not entirely clear. Consider "...ob-microbiota had lower hepatic gluconeogenesis both following acute NC diet and 72%HFD."

→ *We thank the referee for this suggestion which was incorporated in the main-text (page 12), as it follows:*

"...ob-microbiota had lower hepatic gluconeogenesis both following acute NC diet and 72%HFD."

3rd Editorial Decision

08 February 2017

Thank you sending us your revised manuscript. We have now heard back from reviewer #2 who was asked to evaluate your study. As you will see below, the reviewer thinks that all major concerns have now been satisfactorily addressed. S/he raises however some minor issues, all referring to text modifications, which we would ask you to address in a minor revision.

Moreover, before we accept the manuscript, we would ask you to address the following issues:

- According to our journal policy on Data Availability <<http://msb.embopress.org/authorguide#availabilityofpublishedmaterial>> all datasets obtained in the context of the study and integral to the findings reported, should be made available. As such, we would ask you to deposit the metabolomics and metagenomics data in one of the appropriate databases <<http://msb.embopress.org/authorguide#datadeposition>> and include the accession numbers in the Data Availability section. (If you prefer, metabolomics data can be provided as an EV Dataset.) The related section of the Checklist (F-19) should be updated after the data has been made available.

 REFEREE REPORT

Reviewer #2:

The authors have adequately addressed the points raised and provided additional data as requested. The discussion has been significantly improved and is now concise and clearer, as is the language employed throughout the manuscript.

Some minor editing points remain, which are listed below - however, these should not preclude a decision regarding acceptance of the manuscript.

Line 286: smaller not lower

Line 301 and 303: Please check (in red) and (in green). These appear incorrect.

Line 355: the 4th point is "iv"

Line 372 - Would use caution in referring to Raetzsch et al, as this is a study on the acute effects of LPS in lowering glucose production; those authors go on to say that the recovery from acute LPS exposure produces glucose intolerance. In fact, the first few lines of their introduction would argue against your conclusion. You should either omit this line/reference altogether, or rework your speculation on how NOD2 may play a role in improving glucose metabolism in your model.

Line 391: Please clarify that that gene expression 'in the liver' corresponds to smaller adipocytes.

Line 401: reconsider the word 'elaborate'

Line 408-09: this could made clearer; in its current format, these lines suggest that you are invalidating your conclusion from lines 402-07. Please reword so that lines 408-09 are clearly the caveat to your main conclusion (ie, you are acknowledging some of the interesting but not fully-aligned data) without appearing to counter your primary conclusion regarding glyoxylate pathways.

Line 477 - the title of the method is unclear - presumably you did not give insulin orally? Please reword.

Line 800 and 857: space

Line 882: reconsider the words 'well argueded'

3rd Revision - authors' response

15 February 2017

Response to Editor:

- According to our journal policy on Data Availability

<http://msb.embopress.org/authorguide#availabilityofpublishedmaterial> all datasets obtained in the context of the study and integral to the findings reported, should be made available. As such, we would ask you to deposit the metabolomics and metagenomics data in one of the appropriate databases <http://msb.embopress.org/authorguide#datadeposition> and include the accession numbers in the Data Availability section. (If you prefer, metabolomics data can be provided as an EV Dataset.) The related section of the Checklist (F-19) should be updated after the data has been made available.

→ *We are grateful for these suggestions: according to your request metabolomics data have been provided as EV Dataset at the end of the Appendix Figure file; metagenomics data have been deposited to the ENA database. The main text at the section "Data availability" as well as the related section of the Checklist (F-19) have been updated accordingly.*

Point-by-point responses to reviewer

All our corrections to the main text are underlined to help both the editor and the reviewer finding them the easiest way.

Reviewer #2:

The authors have adequately addressed the points raised and provided additional data as requested. The discussion has been significantly improved and is now concise and clearer, as is the language employed throughout the manuscript.

→ *We thank the reviewer for this nice comment. We are grateful to this reviewer for the beneficial inputs provided during the revision process. We felt assisted and we appreciated the positive attitude of both this reviewer and all the Editorial office. We are glad to see how our manuscript has been ameliorated thanks to your kind assistance.*

Some minor editing points remain, which are listed below - however, these should not preclude a decision regarding acceptance of the manuscript.

Line 286: smaller not lower

→ corrected

Line 301 and 303: Please check (in red) and (in green). These appear incorrect.

→ We have checked and there is no error. In fact, to be consistent with all the rest of the figures, for Figure 7 upper panel we put mice inoculated with the HFD-microbiota in green and mice inoculated with ob-microbiota in red; however, the cladograms on the lower panels show mice inoculated with the HFD-microbiota in red and mice inoculated with ob-microbiota in green, as per default of the software. We even contacted the software developer and asked for his help at the time we identified this little (we hope that you consider it as so!) inconsistency, but with no success. Therefore, the text has been slightly modified as it follows (page 13):

- *After the transfer, mice inoculated with the HFD-microbiota and fed a NC showed a gut microbiota deeply different than control mice (Fig. 7B, in green in the upper panel and red in the lower panel). By contrast, the gut microbiota of mice inoculated with the ob-microbiota was almost similar to the one of control mice (Fig. 7B, in red in the upper panel and green in the lower panel).*

Line 355: the 4th point is "iv"

→ corrected

Line 372 - Would use caution in referring to Raetzsch et al, as this is a study on the acute effects of LPS in lowering glucose production; those authors go on to say that the recovery from acute LPS exposure produces glucose intolerance. In fact, the first few lines of their introduction would argue against your conclusion. You should either omit this line/reference altogether, or rework your speculation on how NOD2 may play a role in improving glucose metabolism in your model.

→ We thank the reviewer for this high accurate analysis; to avoid misinterpretations we opt to omit this line/reference altogether, as suggested. The text was slightly modified as it follows (page 16):

- *Being peptidoglycan a NOD2 ligand, we may speculate that NOD2 activation may be implicated in the observed hepatic phenotype.*

Line 391: Please clarify that that gene expression 'in the liver' corresponds to smaller adipocytes.

→ According to this suggestion, we slightly modified the text as it follows (page 17):

- *In fact, Eissing et al. showed that lipogenic enzymes are upregulated in the liver of obese patients (Eissing et al, 2013), and in our study lipogenic enzymes are downregulated in the liver of mice in association with smaller adipocytes.*

Line 401: reconsider the word 'elaborate'

→ We opted for the term 'develop' (page 17)

Line 408-09: this could made clearer; in its current format, these lines suggest that you are invalidating your conclusion from lines 402-07. Please reword so that lines 408-09 are clearly the caveat to your main conclusion (ie, you are acknowledging some of the interesting but not fully-aligned data) without appearing to counter your primary conclusion regarding glyoxylate pathways.

→ We thank the reviewer for this interpretation that we like. The text has been made clearer as it follows (page 17-18):

- *Therefore, targeting microbial genes involved in this pathway may be effective for the control of hepatic function on a NC feeding, but no longer on 72%HFD. ENREF 50In conclusion, our results could open a new debate on the impact of gut microbiota dysbiosis on host metabolism by describing the beneficial effects of the transfer of dysbiotic gut microbiota, principally on the liver. Thus, our new observation may encourage re-examining the causal role of gut microbiota dysbiosis on metabolic diseases.*

Line 477 - the title of the method is unclear - presumably you did not give insulin orally? Please reword.

→ Once again we thank the reviewer for ameliorating the text. We reworded it as it follows (page 20):

- ***Intraperitoneal (IP) Pyruvate-(IPPTT), Glucose-(IPGTT) and Insulin-(IPITT) or oral (O) glucose (OGTT) tolerance tests.***

Line 800 and 857: space

→ *corrected*

Line 882: reconsider the words 'well argued'

→ *These words have been removed. Data are now available (please see above) according to journal policy.*

4th Editorial Decision

20 February 2017

Thank you again for sending us your revised manuscript. We are now satisfied with the modifications made and I am pleased to inform you that your paper has been accepted for publication.

MOLECULAR SYSTEMS BIOLOGY

Corresponding Author Name: Matteo Serino
Manuscript Number: MSB-16-7356